# DIVERSITY BOOSTED LEARNING FOR DOMAIN GENERALIZATION WITH A LARGE NUMBER OF DOMAINS

## ABSTRACT

Machine learning algorithms minimizing the average training loss typically suffer from poor generalization performance. It inspires various works for domain generalization (DG), among which a series of methods work by $O(n^2)$ pairwise domain operations with $n$ domains, where each one is often costly. Moreover, while a common objective in the DG literature is to learn invariant representations against spurious correlations induced by domains, we point out the insufficiency of it and highlight the importance of alleviating spurious correlations caused by *objects*. Based on the observation that diversity helps mitigate spurious correlations, we propose a **D**iversity boosted tw**O**-level sa**M**pl**I**ng framework (`DOMI`) to efficiently sample the most informative ones among a large number of domains and data points. We show that `DOMI` helps train robust models against spurious correlations from both domain-side and object-side, substantially enhancing the performance of five backbone DG algorithms on Rotated MNIST and Rotated Fashion MNIST.

## 1 INTRODUCTION

The effectiveness of machine learning algorithms that minimize the average training loss relies on the IID hypothesis. However, distributional shifts between test and training data are usually inevitable. Under such circumstances, models trained by minimizing the average training loss are prone to sink into spurious correlations. These misleading heuristics only work well on some data distributions but can not be generalized to others that may appear in the test set. In domain generalization (DG) tasks, the data distributions are denoted as different domains. The goal is to learn a model that can generalize well to unseen ones after training on several domains. For example, an image classifier should be able to discriminate the objects whatever the image's background is. While lots of methods have been derived to efficiently achieve this goal and show good performance, there are two main drawbacks.

**Scalability.** With an unprecedented amount of applicable data nowadays, many datasets contain a tremendous amount of domains, or massive data in each domain, or both. For instance, WILDS (Koh et al., 2021) is a curated collection of benchmark datasets representing distribution shifts faced in the wild. Among these datasets, some contain thousands of domains and OGB-MolPCBA (Hu et al., 2020b) contains even more than one hundred thousand. Besides WILDS, DrugOOD (Ji et al., 2022) is an out-of-distribution dataset curator and benchmark for AI-aided drug discovery. Datasets of DrugOOD contain hundreds to tens of thousands of domains. In addition to raw data with multitudinous domains, domain augmentation, leveraged to improve the robustness of models in DG tasks, can also lead to a significant increase in the number of domains. For example, HRM (Liu et al., 2021a) generates heterogeneous domains to help exclude variant features, favoring invariant learning. Under such circumstances, training on the whole dataset in each epoch is computationally prohibitive, especially for methods such as MatchDG (Mahajan et al., 2021) and FISH (Shi et al., 2021b), training by pairwise operations, of which the computational complexity is $O(n^2)$ with $n$ training domains.

**Objective.** Numerous works in the DG field focus entirely on searching **domain-independent** correlations to exclude or alleviate domain-side impacts (Long et al., 2015; Hoffman et al., 2018; Zhao et al., 2018, 2019; Mahajan et al., 2021). We state that this objective is insufficient, and a counterexample is given as follows. We highlight the importance of mitigating spurious correlations caused by the objects for training a robust model.

Suppose our learning task is training a model to distinguish between cats and lions. The composition of the training set is shown in Figure 1, and the domain here refers to the images' backgrounds. In this example, the correlation between features corresponding to the body color of

the objects and class labels is undoubtedly independent of domains. Moreover, it helps get high accuracy in the training set by simply taking the tan objects as lions and the white ones as cats.

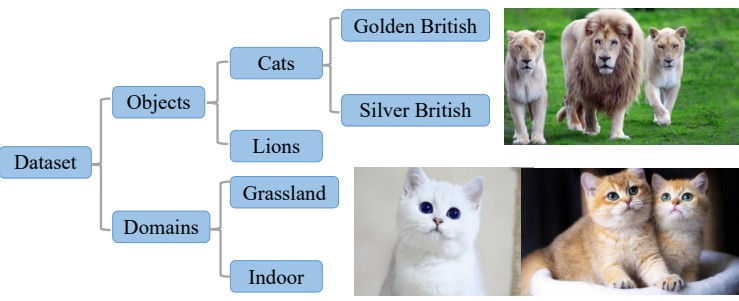

Unfortunately, if this correlation is mistaken for the causal correlation, the model is prone to poor performance once cat breed distribution shifts in the test set.

To tackle these two issues, we propose a diversity boosted two-level sampling framework named `DOMI` with the following major contributions: 1) To our best knowledge, this is the first paper to take im-

Figure 1: The training set of the counterexample. Cats are mainly silver British shorthair (body color of which is silvery white), rarely golden British shorthair (tan), and lions are all tan. As for the background, most lions are on the grassland while most cats are indoors.

pacts from the object side into account for achieving the goal of DG. 2) We propose `DOMI`, a diversity-boosted two-level sampling framework to select the most informative domains and data points for mitigating both domain-side and object-side impacts. 3) We demonstrate that `DOMI` substantially enhances the test accuracy of the backbone DG algorithms on different benchmarks.

## 2 RELATED WORK

**Domain Generalization.** DG aims to learn a model that can generalize well to all domains including unseen ones after training on more than one domains (Blanchard et al., 2011; Wang et al., 2022; Zhou et al., 2021; Shen et al., 2021). Among recent works on domain generalization, Ben-Tal et al. (2013); Duchi et al. (2016) utilize distributionally robust optimization (DRO) to minimize the worst-case loss over potential test distributions instead of the average loss of the training data. Sagawa et al. (2019) propose group DRO to train models against spurious correlations by minimizing the worst-case loss over groups to avoid suffering high losses on some data groups. Zhai et al. (2021) further use distributional and Outlier Robust Optimization (DORO) to address the problem that DRO is sensitive to outliers and thus suffers from poor performance and severe instability when faced with real, large-scale tasks. On the other hand, as Peters et al. (2016) and Rojas-Carulla et al. (2018) state that the predictor should be simultaneously optimal across all domains, (Arjovsky et al., 2019; Javed et al., 2020; Shi et al., 2021a; Ahuja et al., 2020a) leverage Invariant Risk Minimization (IRM) to learn features inducing invariant optimal predictors over training domains. However, Guo et al. (2021); Rosenfeld et al. (2020); Kamath et al. (2021); Ahuja et al. (2020b) point out that works with IRM lack formal guarantees, and IRM does not provably work with non-linear data. Koh et al. (2021) and Gulrajani & Lopez-Paz (2020) present an analysis to demonstrate that IRM fails to generalize well even when faced with some simple data models and fundamentally does not improve over standard ERM. Risk Extrapolation (V-REx) (Krueger et al., 2021) instead hold the view that training risks from different domains should be similar and achieves the goal of DG by matching the risks. Some works explore data augmentations to mix samples from different domains (Wang et al., 2020; Wu et al., 2020) or generate more training domains (Liu et al., 2021a,b) to favor generalization. Another branch of studies assume that data from different domains share some "stable" features whose relationships with the outputs are causal correlations and domain-independent given certain conditions (Long et al., 2015; Hoffman et al., 2018; Zhao et al., 2018, 2019). Among this branch of work, Li et al. (2018c); Ghifary et al. (2016); Hu et al. (2020a) hold the view that causal correlations are independent of domain conditioned on class label, and Muandet et al. (2013) propose DICA to learn representations marginally independent of domain.

**MatchDG.** Mahajan et al. (2021) state that learning representations independent of the domain after conditioning on the class label is insufficient for training a robust model. They propose MatchDG to learn correlations independent of domain conditioned on objects, where objects can be seen as clusters within classes based on similarity. To ensure the learned features are invariant across domains, a term of the distance between each pair of domains is added to the objective to be minimized.

**FISH, MMD, CORAL.** Another line of works promote agreements between gradients with respect to network weights (Koyama & Yamaguchi, 2020; Parascandolo et al., 2020; Rame et al., 2022; Mansilla et al., 2021; Shahtalebi et al., 2021). Among these works, FISH (Shi et al., 2021b) augments

the ERM loss with an auxiliary term of gradient inner product between domains. By minimizing the loss and matching the gradients simultaneously, FISH encourages the optimization paths to be the same for all domains, favoring invariant predictions. MMD (Li et al., 2018b) and CORAL (Sun & Saenko, 2016) are another two matching methods besides MatchDG and FISH. MMD matches the distributions among different domains using Maximum Mean Discrepancy measure. In this way, the learned representation is supposed to be invariant for the training domains. MMD further aligns the matched distribution to an arbitrary prior distribution via adversarial feature learning, aiming to prevent the representation from overfitting to the training domains. Then the learned representation is expected to generalize well on the test domains. CORAL instead matches the second-order statistics of the distributions across domains. Specifically, CORAL concurrently minimizes the ERM loss and the difference in learned feature covariances across domains. As a simple yet effective method, CORAL shows state-of-the-art performance on various tasks for OOD generalization (Gulrajani & Lopez-Paz, 2020).

**DANN.** Besides gradients, some approaches enforce agreements between features and align the features with adversarial methods (Li et al., 2018a; Gong et al., 2016). As one of these approaches, DANN (Ganin et al., 2016) incorporates the structure named domain discriminator to implement adversarial training based on the theory that an ideal classifier for cross-domain shifts should be able to distinguish different classes while cannot learn to identify the domain. DOMI takes use of an inverse version of DANN denoted as invDANN to learn domain-side features and help select the most informative domains.

**DPP.** Determinantal Point Process (DPP) (Kulesza et al., 2012) is a point process that mimics repulsive interactions. A draw from a DPP yields diversified subsets based on a similarity matrix (DPP kernel) of samples to be selected. While it shows powerful performance in selecting heterogeneous data, DPP sampling relies on an eigendecomposition of the DPP kernel, whose cubic complexity is a huge impediment. To address this problem, Li et al. (2016) suggest first construct an approximate probability distribution to the true DPP and then efficiently samples from this approximate distribution. As one choice of diversity sampling, DPP sampling is incorporated into DOMI to help select the most informative domains and data points. It can be replaced with other diversity sampling schemes.

**Discussions.** Although MatchDG, FISH, MMD and CORAL perform well in domain generalization tasks, the matching procedure between domains means their computational complexity is $O(n^2)$ with $n$ training domains. When $n$ is large, it will inevitably slow down the training process. Therefore, we must select the most informative domains from all the training domains. Inspired by Liu et al. (2021a) that heterogeneous training domains help to learn invariant features since more variant features can be excluded, we conduct an analysis of diversity and spurious correlations to further state it.

## 3 DIVERSITY HELPS MITIGATE SPURIOUS CORRELATIONS

Under the circumstance of imbalanced data where specific clusters contain the majority of data points while the others are only a tiny fraction of the training set, there likely to exist an anti-causal path, i.e., a spurious correlation only catching some properties of the large clusters of the data, and algorithms minimizing the average loss like ERM may simply take this correlation as the causal correlation. For example, when faced with chest X-ray dataset (Oakden-Rayner et al., 2020) where many images of patients with pneumothorax contain a thin drain used for treating the disease, a classifier trained by minimizing the average loss can erroneously identify the drains as a predictive feature of the disease. When we sample diverse data, we in fact re-balance them to help mitigate spurious correlations. We verify this observation with a toy example. For the task and dataset mentioned above (Figure 1), we further suppose our featurizer extracts 4 features with a binary value as shown in Table 1. Then

Table 1: Details of the features and the label. $X_1$ to $X_3$ correspond to features of the object and $X_4$ corresponds to features of the domain.

| | $X_1$ : Mane | $X_2$ : Proportion of face | $X_3$ : Body color | $X_4$ : Background | $y$ |
|---|---|---|---|---|---|
| 0 | no mane | short face | white | indoors | cat |
| 1 | have a mane | long face | tan | grassland | lion |

$X_1 + X_2 \geq 1 \Rightarrow y = 1$ is the causal correlation since the proportion of lions' faces is longer than that of cats, and $X_2$ may be wrongly computed to 0 for male lions because of the existence of mane. $X_3 = 1 \Rightarrow y = 1$ is the object-induced spurious correlation (Abbrev. OSC) and $X_4 = 1 \Rightarrow y = 1$ is

the domain-induced spurious correlation (Abbrev. DSC). Details of our simulated dataset is shown in Appendix A.

Suppose we get 6 of these 12 data samples for training where 3 of 6 come from cats and the other 3 are from lions. There are 4 sampling methods denoted as $S_1$ to $S_4$ to be picked: random sampling, diverse sampling with respect to the object features ($X_1$, $X_2$ and $X_3$), diverse sampling with respect to ($X_4$), and diverse sampling with respect to all 4 features. For $S_2$ to $S_4$, we use Manhattan Distance on the corresponding features to measure the similarity between data. After conducting the similarity matrix, we finally use DPP sampling to select data points. Table 2 shows the average training accuracy of OSC and DSC. When the spurious correlations get lower training accuracy, they are less likely to be mistaken for causal correlation, favoring exploration of the causal correlations.

Table 2: We use each sampling method to select 30 batches of data for training, on which the average accuracy of two kinds of spurious correlations is computed. When spurious correlations get lower accuracy during training, they are more likely to be excluded. Here we use DSC and OSC to denote domain-induced spurious correlation and object-induced spurious correlation, respectively.

| Sampling Method | Accuracy of OSC | Accuracy of DSC |
|:---:|:---:|:---:|
| $S_1$ | 0.86 | 0.68 |
| $S_2$ | 0.77 | 0.66 |
| $S_3$ | 0.85 | 0.50 |
| $S_4$ | 0.78 | 0.49 |

We take the data batches sampled by $S_1$ as base-batches. Random sampling preserves the imbalance of data since a data point is more likely to be sampled into a subset when it appears more often in the whole dataset. For base-batches sampled by $S_1$, both OSC and DSC get high accuracy and are thus prone to be wrongly treated as causal correlations. $S_2$ selects diverse data pertaining to object features. Data batches sampled by $S_2$ get lower accuracy for OSC than base-batches, which means $S_2$ reduces the probability of taking OSC as causal correlation. However, data batches sampled by $S_2$ get almost the same result for DSC. $S_3$ selects diverse data on domain-feature $X_4$. For these batches of data, DSC gets lower accuracy than base-batches and is less likely to be taken as causal correlation, while OSC has a similar result. $S_4$ selects data with heterogeneity with regard to all 4 features. Compared to base-batches, the data batches selected by $S_4$ get lower accuracy on both spurious correlations.

## 4 METHODS

Figure 2 shows the sampling procedure of DOMI, a diversity boosted two-level sampling framework.

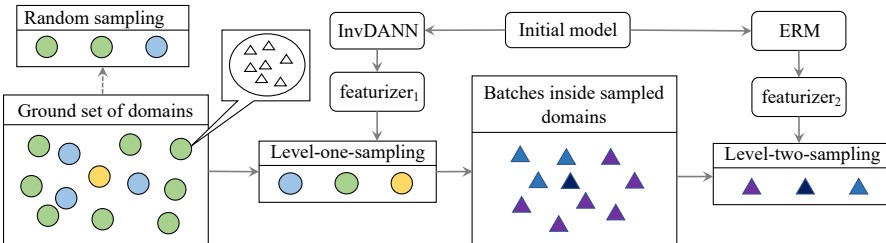

Figure 2: Illustration of the sampling procedure of DOMI. The solid arrow indicates the actual sampling flow, while the dotted arrow is only used to demonstrate the difference between random sampling and DOMI.

### 4.1 DIVERSITY BOOSTED SAMPLING FRAMEWORK

**Preliminaries.** Consider the universal set of domains $\mathcal{D}$, where each domain $d \in \mathcal{D}$ corresponds to a distribution $P_d$ over $\mathcal{X} \times \mathcal{Y}$, with $\mathcal{X}$ being the space of inputs and $\mathcal{Y}$ that of outputs. Our goal is to find a predictor $f : \mathcal{X} \to \widehat{\mathcal{Y}}$ while we can only access the domains in $\mathcal{D}_{tr}$ where $\mathcal{D}_{tr} \subset \mathcal{D}$. We measure the quality of a prediction with a loss function $\ell : \widehat{\mathcal{Y}} \times \mathcal{Y} \to R_{\geq 0}$; and the quality of a predictor by its population loss on domain $d \in \mathcal{D}$, given by $\mathcal{L}_d(f) := E_{(x,y) \sim P_d}[\ell(f(x), y)]$.

**Definition 1** (Correlation). *We define a correlation as a predictor $f = \omega \circ \phi$ where $\phi : \mathcal{X} \to \mathcal{Z}$ is a data representation and $\omega^* : \mathcal{Z} \to \mathcal{Y}$ is a classifier. The causal correlation $f^*$ satisfies $\phi^*$ elicits a invariant predictor (Arjovsky et al., 2019) on $\mathcal{D}$: $\omega^*$ simultaneously optimal for all domains, i.e., $\forall d \in \mathcal{D}, \omega^* \in argmin_{\omega:\mathcal{Z}\to\mathcal{Y}}\mathcal{L}_d(\omega \circ \phi^*)$.*

Notably, Definition 1 requires that $\phi$ and $\omega$ are unrestricted in the space of all (measurable) functions. However, we learn $\phi$ and $\omega$ being restricted to only access domains in $\mathcal{D}_{tr}$, a small subset of $\mathcal{D}$. For this to be feasible, it is natural to add a restriction that $\phi \in \Phi$ and $\omega \in \Omega$ for suitable classes $\Phi$ of functions mapping $\mathcal{X} \to \mathcal{Z}$ and W of functions mapping $\mathcal{Z} \to \widehat{\mathcal{Y}}$.

**Assumption 1.** $argmin_{\omega:\mathcal{Z}\to\mathcal{Y}}\mathcal{L}_d(\omega \circ \phi) = \{\omega | \mathcal{L}_d(\omega \circ \phi) \leq \delta\}$, where $\delta > 0$ is a constant.

**Definition 2.** *Consider a domain set $\mathcal{D}_s$, on which the set of invariant predictors, $\mathcal{I}(\mathcal{D}_s)$, is the set of all predictors $f$ satisfies following:* • $f = \omega \circ \phi$ with $(\omega, \phi) \in \Omega \times \Phi$; • *for all* $d \in \mathcal{D}_s$, $\omega \in argmin_{\bar{\omega}:\mathcal{Z}\to\mathcal{Y}}\mathcal{L}_d(\bar{\omega} \circ \phi)$.

**Lemma 4.1.** *Based on Definition 1 and Definition 2, we can trivially derive: for any nonempty set $\bar{\mathcal{D}} \subseteq \mathcal{D}$, $f^* \in \mathcal{I}(\bar{\mathcal{D}})$.*

**Definition 3** (Diversity). *We use Integral Probability Metric (Müller, 1997) to measure the diversity between domains. For domain $d$ and $\bar{d}$, the diversity is defined as:*

$$Div(P_d, P_{\bar{d}}) = Div(P_d, P_{\bar{d}}, \mathcal{G}) = \sup_{g \in \mathcal{G}} |E_{P_d}[g(x,y)] - E_{P_{\bar{d}}}[g(x,y)]|$$

*Where $\mathcal{G}$ is a class of bounded functions. When we let $g(x,y) = \ell(f(x), y)$ and $\mathcal{G} = \mathcal{F} = \Omega \times \Phi$, the diversity is:*

$$Div(P_d, P_{\bar{d}}) = Div(P_d, P_{\bar{d}}, \Omega, \Phi) = \sup_{\omega \in \Omega, \phi \in \Phi} |E_{P_d}[\ell(\omega \circ \phi(x), y)] - E_{P_{\bar{d}}}[\ell(\omega \circ \phi(x), y)]|$$

$$= Div(P_d, P_{\bar{d}}, \mathcal{F}) = \sup_{f \in \mathcal{F}} |\mathcal{L}_d(f) - \mathcal{L}_{\bar{d}}(f)|$$

Consider we have a domain set $\mathcal{D}_k = \{d_1, d_2..., d_k\}$ and the corresponding $\mathcal{I}(\mathcal{D}_k) = \{f_1, f_2..., f_m\}$. And now we get one more domain $d_{k+1}$ to form $\mathcal{D}_{k+1}$. According to Lemma 4.1, the causal correlation $f^* \in \mathcal{I}(\mathcal{D}_{k+1})$, so a informative domain $d_{k+1}$ which helps exclude spurious correlations leads to $|\mathcal{I}(\mathcal{D}_{k+1})|] < m$.

**Proposition 1** (Diverse domains help exclude spurious correlations.). *If $d_{k+1}$ satisfies that:* $\max\limits_{d \in \mathcal{D}_k, f_i \in \mathcal{I}(\mathcal{D}_k)} Div(d_{k+1}, d) + \mathcal{L}_d(f_i) \leq \delta$, *then $\mathcal{I}(\mathcal{D}_k) = \mathcal{I}(\mathcal{D}_{k+1})$.*

*Proof.* Without loss of generality, we first conduct analysis on $f_t$ of $\mathcal{I}(\mathcal{D}_k)$. For $f_t$:

$$\max_{d \in \mathcal{D}_k} |\mathcal{L}_{d_{k+1}}(f_t) - \mathcal{L}_d(f_t)| + \mathcal{L}_d(f_t) \leq \max_{d \in \mathcal{D}_k} Div(d_{k+1}, d) + \mathcal{L}_d(f_t)$$

$$\max_{d \in \mathcal{D}_k} Div(d_{k+1}, d) + \mathcal{L}_d(f_t) \leq \max_{d \in \mathcal{D}_k, f_i \in \mathcal{I}(\mathcal{D}_k)} Div(d_{k+1}, d) + \mathcal{L}_d(f_i) \leq \delta$$

When

• $\mathcal{L}_{d_{k+1}}(f_t) - \mathcal{L}_d(f_t) < 0$ :
$\mathcal{L}_{d_{k+1}}(f_t) < \mathcal{L}_d(f_t) \leq \delta$
• $\mathcal{L}_{d_{k+1}}(f_t) - \mathcal{L}_d(f_t) \geq 0$ :
$\mathcal{L}_{d_{k+1}}(f_t) = \max_{d \in \mathcal{D}_k} \mathcal{L}_{d_{k+1}}(f_t) - \mathcal{L}_d(f_t) + \mathcal{L}_d(f_t) = \max_{d \in \mathcal{D}_k} |\mathcal{L}_{d_{k+1}}(f_t) - \mathcal{L}_d(f_t)| + \mathcal{L}_d(f_t) \leq \delta$

$\mathcal{L}_{d_{k+1}}(f_t) \leq \delta$, we get $f_t \in \mathcal{I}(\mathcal{D}_{k+1})$ for any $t \in \{1, 2..., m\}$, thus $\mathcal{I}(\mathcal{D}_k) = \mathcal{I}(\mathcal{D}_{k+1})$ $\qquad\square$

### 4.1.1 DIVERSITY SAMPLING METHOD

As Proposition 1 states that diversity helps mitigate spurious correlations, `DOMI` is a diversity boosted sampling framework and the sampling scheme to obtain a heterogeneous subset is a critical part of `DOMI`. Determinantal Point Process (DPP) sampling is a powerful diversity sampling method. Based on the similarity matrix between the samples, a draw from a DPP yields diversified subsets. Thus we incorporate DPP sampling into `DOMI`. As one option for the diversity sampling method in `DOMI`, DPP sampling can also be substituted with other sampling methods.

### 4.2 LEVEL-ONE-SAMPLING

In the level-one-sampling, we select diverse domains to help mitigate domain-induced spurious correlations. Since we aim to sample diverse domains, we have to learn about the domains. We propose an inverse version of DANN denoted as `invDANN` to train a model to capture the domain information.

### 4.2.1 INVDANN

Domain-Adversarial Neural Networks (DANN) proposed by (Ganin et al., 2016) is composed by a featurizer, a classifier and a discriminator. The featurizer extracts features of data samples, the classifier learns to classify class labels of data, and the discriminator learns to discriminate domains. Since DANN aims to obtain a model that can not differentiate domains to ensure the featurizer captures domain-independent features, the discriminator is connected to the the featurizer via a gradient reversal layer that multiplies the gradient by a certain negative constant during backpropagation. Gradient reversal ensures that the feature distributions over the two domains are made similar, thus resulting in domain-independent features. Using the architecture of DANN, we let the classifier learn to classify domain labels of data while the discriminator learns to discriminate class labels. As an inverse version of DANN, `invDANN` trains a model that can classify domains while not distinguishing class labels. Thus we can get a featurizer extracting only domain-side features.

### 4.2.2 SAMPLING PROCEDURE

In the level-one-sampling of `DOMI`, we first use `invDANN` to train a featurizer. As mentioned in Section 4.2.1, the featurizer only extracts domain-side features. Then we use the featurizer to capture the information of domains and construct a similarity matrix between them. Based on the similarity matrix, DPP sampling selects the diverse domains.

## 4.3 LEVEL-TWO-SAMPLING

**Observation 1.** *Excluding domain-induced spurious correlations is insufficient for learning a robust model.*

Mahajan et al. (2021) have proposed that correlations independent of domain conditional on class $(\Phi(x) \perp\!\!\!\perp D|Y)$ are not necessarily causal correlations if $P(\dot{x}|Y)$ changes across domains. Here $\Phi(x)$ is a featurizer to extract features and $\dot{x}$ represents the causal features. We now further propose that the condition is insufficient even if $\dot{x}$ is consistent across domains. A correlation incorporating features entirely from the objects can still be a spurious correlation. Figure 3 shows a structural causal model (SCM) that describes the data-generating process for the domain generalization task. The SCM divides data into two parts: domain-side and object-side. $\overline{x}$ of domain-side is the reason for domain-induced spurious correlations. For object-side, feature is further divided into $\dot{x}$ and $\widehat{x}$ where $\widehat{x}$ is the reason for object-induced spurious correlations, just like the body color of objects in the toy example. The three parts together make up the observed data. Thus even if we exclude all the domain-induced spurious correlations, i.e., entirely remove the effect from $\overline{x}$, we may still obtain object-induced spurious correlations resulting from $\widehat{x}$.

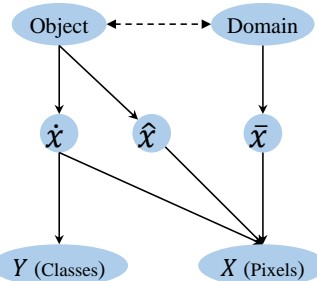

Figure 3: The Structural Causal Model for the data-generating process with a node $\hat{x}$ leading to object-induced spurious correlations.

### 4.3.1 SAMPLING PROCEDURE

As observation 1 shows that excluding only domain-induced spurious correlations is insufficient, we select diverse data batches among the selected domains to help mitigate object-induced spurious correlations in the level-two-sampling. In the level-two-sampling, since we do not have available labels just like domain labels in the level-one-sampling, it is infeasible to utilize `invDANN` again to train a featurizer. So we instead use an ERM model since ERM is prone to taking shortcuts and learning spurious correlations (Zhang et al., 2022). Zhang et al. (2022) also leverage an ERM model to infer the spurious attributes in the unsupervised DG field. Moreover, since domains attained by the level-one-sampling contain diverse data with respect to the domain side, ERM can avert learning domain-induced spurious correlations. Combining these two, the ERM model is prone to relying on object-induced spurious correlations and thus can extract their information. Then a similarity matrix between data batches is constructed with respect to this information. Based on which DPP sampling selects the data batches helping exclude object-induced spurious correlations.

## 4.4 DOMI

We present `DOMI` in Algorithm 1. Combining the two levels, `DOMI` finally gets a subset of the dataset to tackle the issue of scalability under the setting of tremendous domains and training on which helps obtain robust models against impacts from both domain-side and object-side.

---

**Algorithm 1:** Sampling Procedure of `DOMI`

---

**Input:** The whole training dataset:
$$T = [\{(x_i^d, y_i^d)\}_{i=1}^{n_d} \text{ for } d \in \mathbf{D}]$$
the proportion of domains ($\beta$) and batches ($\delta$) to be sampled

1 **Level-one-sampling**
2 Train an `invDANN` featurizer $f_{\bar{\theta}}$ on a randomly sampled subset of $T$ ;
3 **for** $d$ *in* $\mathbf{D}$ **do**
4    $\text{feat}_d \leftarrow \vec{0}$ ;
5    **for** $i$ *from 1 to* $n_d$ **do**
6      $\text{feat}_d \leftarrow \text{feat}_d + f_{\bar{\theta}}(x_i^d)$ ;
7    $\text{feat}_d \leftarrow \text{feat}_d \cdot \frac{1}{n_d}$ ;
8 Initialize similarity matrix $L_d = 0_{|\mathbf{D}| \times |\mathbf{D}|}$ ;

9 **for** $d_i$ *in* $\mathbf{D}$ **do**
10    **for** $d_j$ *in* $\mathbf{D}$ **do**
11      $L_d[i][j] = \|\text{feat}_{d_i} - \text{feat}_{d_j}\|_2$ ;
12 Obtain $\Omega = \text{DPP}(L_d, \beta \cdot |\mathbf{D}|) = [\{(x_i^d, y_i^d)\}_{i=1}^{n_d}$ for $d \in D], (D \subset \mathbf{D}, |D| = \beta \cdot |\mathbf{D}|)$ ;
13 **Level-two-sampling**
14 Divide $\Omega$ into $R = [\{(x_i^b, y_i^b)\}_{i=1}^{n}$ for $b \in \mathbf{B}]$ ;
15 Train an ERM featurizer $f_{\hat{\theta}}$ on $R$;
16 **for** $b$ *in* $\mathbf{B}$ **do**
17    Compute $\text{feat}_b$ in the same way as computing $\text{feat}_d$ in **Level-one-sampling**;
18 Computing similarity matrix $L_b$ ;
19 Return $S = \text{DPP}(L_b, \delta \cdot |\mathbf{B}|)$ ;

---

## 5 EXPERIMENTS

We have investigated the performance of `DOMI` with different backbone DG algorithms on four benchmarks, which show that `DOMI` can help substantially achieve higher test accuracy. We also conduct experiments on iwildcam. Due to space constraints, the results and analysis are listed in Appendix B.3. The experimental settings and results are shown as follows.

### 5.1 CONFIGURATIONS

**Backbones.** We take MatchDG (Mahajan et al., 2021), FISH (Shi et al., 2021b), CORAL (Sun & Saenko, 2016), MMD (Li et al., 2018b) and DANN (Ganin et al., 2016) as backbone algorithms. The former four algorithms work by pairwise domain operations, leading to $O(n^2)$ computational complexity with $n$ domains and thus prohibitive to be scaled to DG tasks with multitudinous domains. It is essential for them to sample the most informative domains. We further incorporate DANN as one of the backbone algorithms since `DOMI` can not only efficiently select domains by its first level of sampling but can help deal with circumstances where each domain contains massive data by the second level of sampling.

**Baselines.** For each one of the backbone algorithms, we set the baseline as training on domains selected by the random sampling scheme and denote it as $level_0$, compared to the level-one-sampling of `DOMI` and the full version of `DOMI` represented as $level_1$ and $level_2$, respectively. We sample 5 domains, i.e., $\beta = 5/61$ for training on Rotated MNIST and Rotated Fashion MNIST. The proportion of minibatches selected in level-two-sampling ($\delta$) is a hyperparameter valued from 0 to 1. When $\delta$ equals 1, $level_2$ shrinks to $level_1$. Within each backbone algorithm, we keep factors including learning rate, batch size, choice of optimizer and model architecture the same for $level_0$, $level_1$ and $level_2$ to highlight the effect of different sampling schemes. It's worth noting that we do no comparison between the backbone algorithms since we do not conduct meticulous hyperparameter tuning for them.

**Model selection.** During training, we use a validation set to measure the model's performance. The test accuracy of the model is updated after an epoch if it shows better validating performance. That is, we save the model with the highest validation accuracy after the training procedure, obtain its test accuracy and report results. For Rotated MNIST and Rotated Fashion MNIST, data from only source domains (rotation degree is from 15 ° to 75 °) are used to form the validation set.

### 5.2 EXPERIMENTS ON MNISTS.

We first conduct experiments with five backbone DG algorithms on two simulated benchmarks (Rotated MNIST and Rotated Fashion MNIST).

#### 5.2.1 ROTATED MNIST AND ROTATED FASHION MNIST

To satisfy the setting of a large number of domains, we extend the original simulated benchmarks on MNIST and Fashion MNIST by Piratla et al. (2020) from rotating images 15° through 75° in intervals of 15° to intervals of 1° in the training set, i.e., 61 domains in total. And we get test accuracy on the test set which rotates images either 0° or 90°. Moreover, while the original datasets rotate the same images for different degrees, we extend them to fit the real cases in DG tasks. We generate indices

Table 3: Average test accuracy of five algorithms. We repeat the experiment for 5 times on FISH and 20 times on the other algorithms with random seeds.

| Dataset | Sampling scheme | DANN | MatchDG | FISH | MMD | CORAL |
|---|---|---|---|---|---|---|
| Rotated MNIST | $level_0$ | 74.5 | 81.5 | 65.2 | 84.2 | 85.6 |
| | $level_1$ | 76.5 ↑2.0 | 83.6 ↑2.1 | 66.5 ↑1.3 | 87.2 ↑3.0 | 89.2 ↑3.6 |
| | $level_2$ | **78.6** ↑4.1 | **84.2** ↑2.7 | **66.6** ↑1.4 | **87.7** ↑3.5 | **89.6** ↑4.0 |
| Rotated Fashion MNIST | $level_0$ | 40.3 | 38.2 | 33.2 | 39.0 | 38.7 |
| | $level_1$ | 42.8 ↑2.5 | 39.7 ↑1.5 | 34.5 ↑1.3 | 41.8 ↑2.8 | 40.8 ↑2.1 |
| | $level_2$ | **43.5** ↑3.2 | **40.7** ↑2.5 | **35.8** ↑2.6 | **42.8** ↑3.8 | **42.1** ↑3.4 |

using different random seeds to select images from MNIST and Fashion MNIST for each domain before rotating. Appendix C gives examples to show how spurious correlations can occur in MNISTs.

### 5.2.2 EMPIRICAL RESULTS AND ANALYSIS ON MNISTS

Table 3 shows the empirical results and we make the following observations:

**Strong performance across datasets and algorithms.** Considering results of 5 backbone DG algorithms on MNISTs, $level_1$ gives constant and apparent improvement over $level_0$. While $level_2$ may lead to slower growth in accuracy at the initial part of training as shown in Figure 4 because of using a smaller number of minibatches, it keeps outperforming $level_1$ and $level_0$ at later epochs.

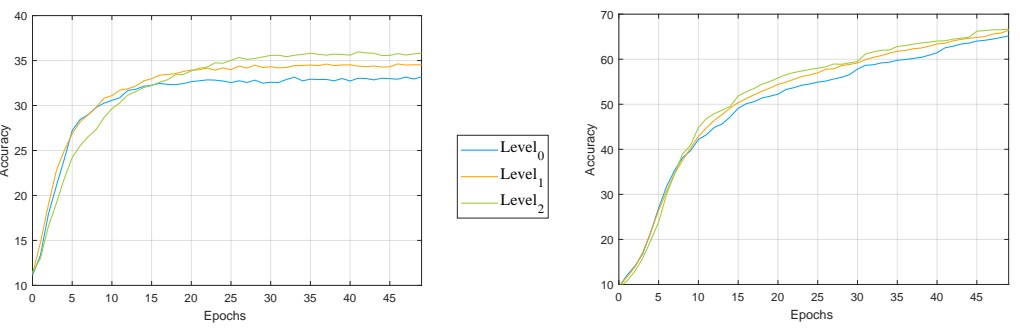

(a) Rotated Fashion MNIST          (b) Rotated MNIST

Figure 4: Average test accuracy of 5 experiments with random seeds during 50 epochs under different sampling schemes of FISH.

**The gap between test accuracy and maximal accuracy.** During training we observe that the test accuracy first rises to the peak value and then begins to decline along with the increase of validation accuracy. This reduction indicates a certain degree of overfitting to spurious correlations. Thus we further record the peak value of the test accuracy in each experiment and denote it as maximal accuracy. The distribution of test accuracy and maximal accuracy on MatchDG under different sampling schemes is shown in Figure 5. While the test accuracy of $level_0$ scatters, that of $level_2$ centers and $level_2$ shrinks the gap between the test accuracy and the maximal accuracy.

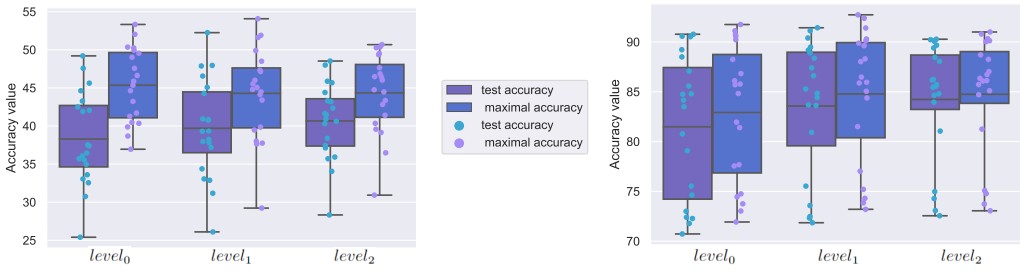

(a) Rotated Fashion MNIST          (b) Rotated MNIST

Figure 5: Boxplot of test accuracy and maximal accuracy among 20 repeated experiments with random seeds leveraging different sampling levels on MNISTs. Among training epochs, the test accuracy rises to the peak value and then declines with the increase of validation accuracy. In this figure, maximal accuracy represents the peak value. Each tiny circle represents one time of experiment, of which the vertical location corresponds to the accuracy value. The horizontal line inside each box indicates the mean value.

**The choice of $\delta$.** A smaller $\delta$ helps efficiently mitigate strong object-induced spurious correlations and speed up training, but when the impact from object-side is weak, a small $\delta$ leads to a waste of training data. In the experiment we observe that a relatively small $\delta$ is more beneficial for Rotated Fashion MNIST while a large $\delta$ works better on Rotated MNIST. Figure 6 shows the results of different $\delta$.

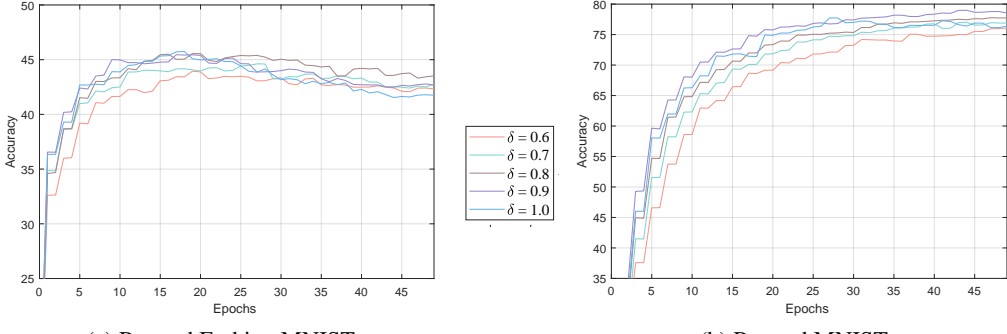

(a) Rotated Fashion MNIST

(b) Rotated MNIST

Figure 6: Average test accuracy of 20 experiments with random seeds during 50 epochs with different $\delta$ on MNISTs of DANN. $\delta = 1.0$ corresponds to `DOMI` with only level one.

### 5.3    EXPERIMENTS ON CIFARS.

We then extend our experiments to two more challenging benchmarks (CIFAR10-C and CIFAR100-C) with three backbone DG algorithms.

### 5.3.1    CIFAR10-C AND CIFAR100-C.

To inspect the robustness of Neural Networks to common corruptions and perturbations, Hendrycks & Dietterich (2019) add noises such as frost and fog effects to images by 19 corruption functions with different severity. In our experiments, we modify the original CIFAR10-C, CIFAR100-C in (Hendrycks & Dietterich, 2019). After generating indices using different random seeds to select images from CIFAR10 and CIFAR100, we use 2 of 19 functions to transform images as test domain data. As for training domains, we leverage the left 17 corruption functions where each function holds three severity: light, medium, and hard, i.e., there are 51 domains in the training set.

### 5.3.2    EMPIRICAL RESULTS AND ANALYSIS ON CIFARS

The empirical results in Table 4 show the improvement of `DOMI` across datasets and algorithms. Detailed settings and more results are listed in Appendix B.2. We further conduct experiments

Table 4: Average test accuracy of three algorithms. We repeat the experiment for 20 times with random seeds.

| Dataset | Sampling scheme | DANN | MMD | CORAL |
|---|---|---|---|---|
| CIFAR10-C | $level_0$ | 63.4 | 65.0 | 68.5 |
|  | $level_1$ | 64.2 ↑0.8 | 66.5 ↑1.5 | 70.1 ↑1.6 |
|  | $level_2$ | **64.6** ↑1.2 | **66.9** ↑1.9 | **70.7** ↑2.2 |
| CIFAR100-C | $level_0$ | 33.9 | 33.8 | 35.1 |
|  | $level_1$ | 34.9 ↑1.0 | 35.2 ↑1.4 | 36.3 ↑1.2 |
|  | $level_2$ | **35.3** ↑1.4 | **35.7** ↑1.9 | **37.0** ↑1.9 |

to compare sampling and non-sampling lines. The experimental settings and results are shown in Appendix B.4. Combined with Appendix B.3, it would be a significant future work to tackle two issues: extremely imbalanced data; computational overhead for algorithms that need to do sampling for multi-times. Moreover, parameter sharing between the trained model and the model used to sample is likely to be a practical means of reducing computational overhead.

## 6    CONCLUSION

Under the setting of a large number of domains and domains with massive data points, we propose a diversity-boosted two-level sampling algorithm named `DOMI` to help sample the most informative subset of dataset. Empirical results show that `DOMI` substantially enhances the out-of-domain accuracy and gets robust models against spurious correlations from both domain-side and object-side.

## ETHICS STATEMENT

This study does not involve any of the following: human subjects, practices to dataset releases, potentially harmful insights, methodologies and applications, potential conflicts of interest and sponsorship, discrimination/bias/fairness concerns, privacy and security issues, legal compliance, and research integrity issues.

## REPRODUCIBILITY STATEMENT

To ensure the reproducibility of our empirical results, we present the detailed experimental settings in Appendix B.1 in addition to the main text. Besides, we will further provide the source codes for reproducing results in our paper.

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

# Appendix of DOMI

CONTENTS

## A  THE SIMULATED DATASET

Table 5: The simulated dataset of the toy example. From these 12 data points we sample 6 for training.

|       | $D_1$ | $D_2$ | $D_3$ | $D_4$ | $D_5$ | $D_6$ | $D_7$ | $D_8$ | $D_9$ | $D_{10}$ | $D_{11}$ | $D_{12}$ |
|-------|-------|-------|-------|-------|-------|-------|-------|-------|-------|----------|----------|----------|
| $X_1$ | 0     | 0     | 0     | 0     | 0     | 0     | 0     | 0     | 0     | 1        | 1        | 1        |
| $X_2$ | 0     | 0     | 0     | 0     | 0     | 0     | 1     | 1     | 1     | 0        | 1        | 1        |
| $X_3$ | 0     | 0     | 0     | 0     | 1     | 1     | 1     | 1     | 1     | 1        | 1        | 1        |
| $X_4$ | 0     | 0     | 0     | 1     | 0     | 1     | 1     | 1     | 0     | 0        | 1        | 1        |
| $Y$   | 0     | 0     | 0     | 0     | 0     | 0     | 1     | 1     | 1     | 1        | 1        | 1        |

## B  EXPERIMENTS

### B.1  SETTINGS AND RESULTS ON MNISTS

For DANN, the training epochs are set to be 50. MatchDG is a two-phase method, and in our experiment we set 30 epochs of training for phase 1 and 25 epochs for phase 2. The training epochs of FISH are set to be 5. Each epoch contains 300 iterations and we observe test accuracy every 30 iterations. And in Figure 4 we slightly abuse epoch to mean the time we obtain test accuracy. Unlike MatchDG and DANN, fish needs to sample domains in each iteration instead of training on one list of domains. Sampling domains in each iteration will result in great computational overhead compared to randomly sampling. Thus we just sample 30 domain lists containing diverse domains using level-one-sampling of DOMI and repeatedly train the model on these domain lists(one list for one iteration) for $level_1$. As for $level_2$, we further utilize level-two-sampling to sample data batches of each domain in the domain lists for training. The former 3 DG algorithms utilize SGD optimizer with learning rate 0.01, weight decay 5e-4 and momentum 0.9. The training epochs of MMD and CORAL are set as 30. These two algorithms leverage Adam optimizer with learning rate 0.001 and weight decay 0. All five algorithms use Resnet18 model. Figure 7 show test accuracy and maximal accuracy among 20 times of repeated experiments with random seeds leveraging different sampling levels on Rotated Fashion MNIST and Rotated MNIST. Among training epochs, the test accuracy rises to the peak value and then decline along with the increase of validation accuracy. In this figure maximal accuracy represents the peak value. Each tiny circle represents one times of experiment, of which the vertical location corresponds to the accuracy value. The horizontal line inside each box indicates the mean value.

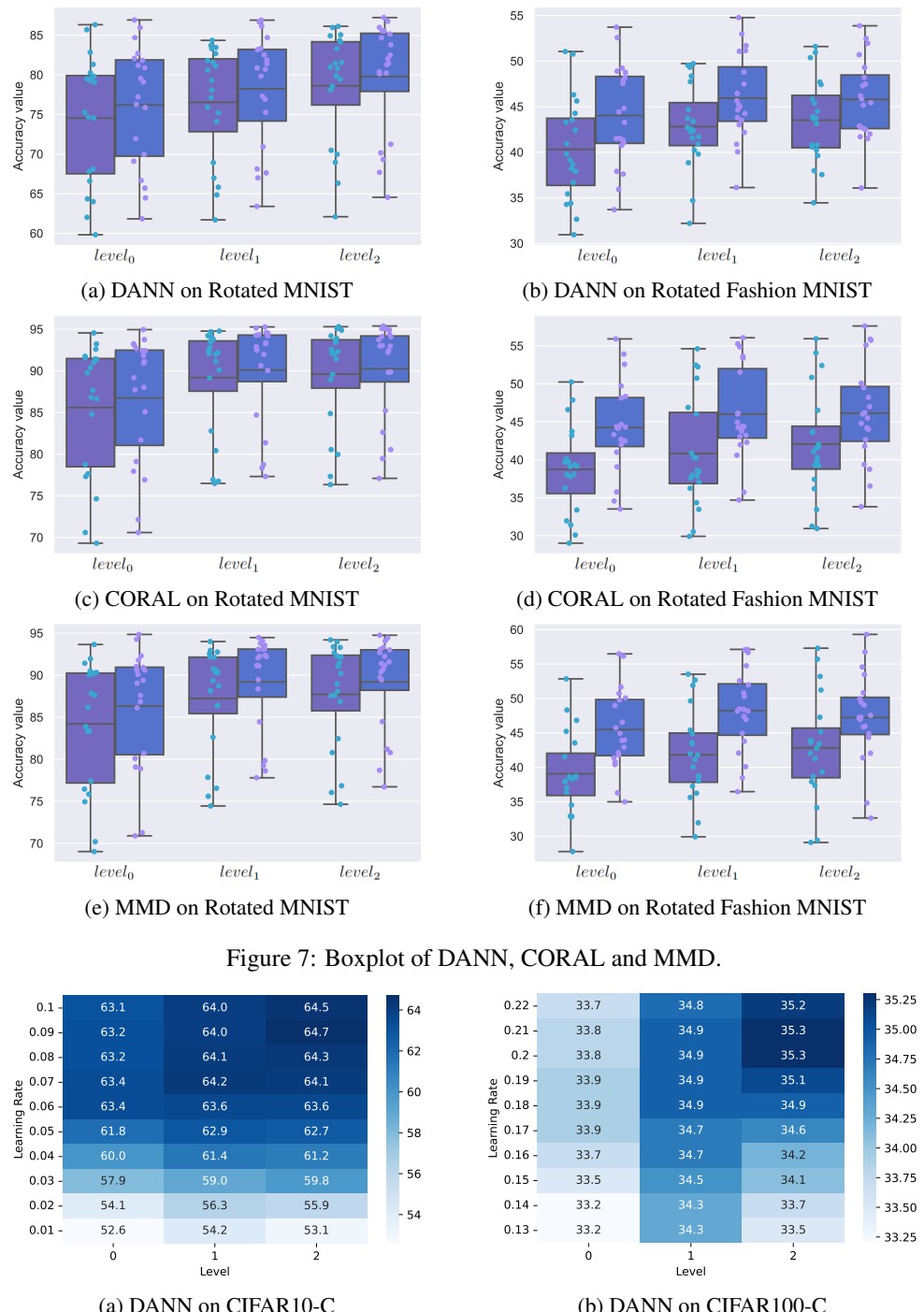

Figure 7: Boxplot of DANN, CORAL and MMD.

(a) DANN on CIFAR10-C

(b) DANN on CIFAR100-C

Figure 8: Average test accuracy of 20 experiments with random seeds using different learning rates under three sampling schemes of DANN.

## B.2 SETTING AND RESULTS ON CIFARS

The training epochs are set to be 60 for DANN and 30 for MMD and CORAL. DANN utilize SGD optimizer with weight decay 5e-4 and momentum 0.9. Figure 8 shows the results of DANN using different learning rates. MMD and CORAL leverage Adam optimizer with learning rate 0.001 and weight decay 0. The loss of MMD and CORAL is the sum of a ERM-loss term and a term of weighted distance between domains. The weight is set as 1.0 on two CIFAR datasets for CORAL while 1.0 on CIFAR10-C and 0.2 on CIFAR100-C for MMD. All three algorithms use Resnet18

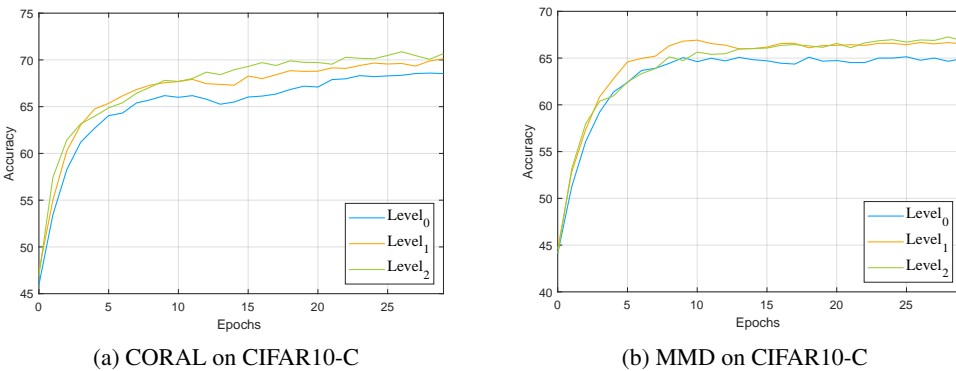

(a) CORAL on CIFAR10-C  (b) MMD on CIFAR10-C

Figure 9: Average test accuracy of 20 experiments with random seeds during 30 epochs leveraging different sampling levels.

Table 6: Macro F1 score of FISH on iwildcam under three sampling schemes.

|  | $level_0$ | $level_1$ | $level_2$ |
|---|---|---|---|
| Iwildcam | 22.0 | 22.8 | **23.4** |

model, hyperparameters $\delta = 0.9$, $\beta = 5/51$ on two CIFAR datasets. Figure 9 show average test accuracy of 20 experiments with random seeds during 30 epochs leveraging different sampling levels on CIFAR10-C.

### B.3 EXPERIMENTS ON IWILDCAM

WILDS (Koh et al., 2021) is a curated collection of benchmark datasets representing distribution shifts faced in the wild. As one dataset in WILDS, iwildcam contains photos of wild animals and 324 different camera traps are taken as domains. The data of iwildcam is extremely unbalanced, while part of the domains contain less than 20 photos, some domains contain over 2000 ones. In the original experiments of Shi et al. (2021b), iwildcam is divided into batches in each domain. FISH samples 10 batches from different domains for training in each iteration. The sampling probability of one batch in a domain is proportional to the number of batches left in this domain. This sampling scheme is taken as $level_0$ here and we refer to the result of (Shi et al., 2021b). In each iteration, $level_1$ samples 10 batches based on DPP using invDANN, $level_2$ first samples 10 batches in the level-one-sampling and among them selects 6 batches in the level-two-sampling. Under the same setting in the original experiments, the results on iwildcam of FISH are shown in Table 6 .

Although DOMI gets higher Macro F1 score, it leads to a much larger computational overhead since it needs to do sampling in each iteration. Moreover, for DANN and MatchDG, Macro F1 of diverse domains may be significantly lower than randomly sampled domains because of the unbalanced data, i.e., the diverse domains may contain much fewer data compared to the randomly sampled domains.

### B.4 WHY SAMPLING?

We conduct this experiment on CIFAR10-C and CIFAR100-C with two backbone algorithms: DANN and CORAL. We set training on all 51 domains in each epoch as the base line.

**One-shot sampling & non-sampling.** We first do comparison between one-shot sampling lines., i.e., sample a subset of domains and data for training for all epochs, and non-sampling line on accuracy and wall-clock time required. Specifically, we supplement the spent wall-clock time to Table 4 and add experiments on non-sampling line with settings as follows: the training epochs are set to be 15 for DANN and 10 for CORAL; the deployed Neural Network is Resnet18; the experiment with DANN utilize SGD optimizer with learning rate 0.2, weight decay 5e-4 and momentum 0.9; the experiment with CORAL leverage Adam optimizer with learning rate 0.001 and weight decay 0. Table 7 shows the accuracy and wall-clock time of each line.

Table 7: Accuracy(%) and wall-clock time(seconds).

| Dataset | Backbone Algorithm | Sampling Scheme | Accuracy | Wall-clock Time |
|---|---|---|---|---|
| CIFAR10-C | DANN | $level_0$ | 63.4 | 916(0.22×) |
| | | $level_1$ | 64.2 | 1325(0.32×) |
| | | $level_2$ | 64.6 | 1523(0.37×) |
| | | Non-sampling | 89.3 | 4164(1.00×) |
| | CORAL | $level_0$ | 68.5 | 461(0.05×) |
| | | $level_1$ | 70.1 | 863(0.09×) |
| | | $level_2$ | 70.7 | 1074(0.11×) |
| | | Non-sampling | 94.2 | 10152(1.00×) |
| CIFAR100-C | DANN | $level_0$ | 33.9 | 945(0.22×) |
| | | $level_1$ | 34.9 | 1427(0.33×) |
| | | $level_2$ | 35.3 | 1664(0.38×) |
| | | Non-sampling | 71.0 | 4368(1.00×) |
| | CORAL | $level_0$ | 35.2 | 496(0.04×) |
| | | $level_1$ | 36.3 | 902(0.08×) |
| | | $level_2$ | 37.0 | 1158(0.10×) |
| | | Non-sampling | 85.3 | 11361 (1.00×) |

Table 8: Mean accuracy(%) and wall-clock time(seconds). We repeat the experiment of each line for 3 times with different random seeds.

| Dataset | Backbone Algorithm | Sampling Scheme | Accuracy | Wall-clock Time |
|---|---|---|---|---|
| CIFAR10-C | DANN | $level_0$ | 82.2 | 923(0.22×) |
| | | $level_1$ | 83.5 | 1531(0.37×) |
| | | $level_2$ | 83.9 | 6712(1.61×) |
| | | Non-sampling | 89.3 | 4164(1.00×) |
| | CORAL | $level_0$ | 87.8 | 461(0.05×) |
| | | $level_1$ | 89.2 | 1004(0.10×) |
| | | $level_2$ | 89.7 | 6175(0.61×) |
| | | Non-sampling | 94.2 | 10152(1.00×) |
| CIFAR100-C | DANN | $level_0$ | 61.7 | 945(0.22×) |
| | | $level_1$ | 62.9 | 1674(0.38×) |
| | | $level_2$ | 63.4 | 6864(1.57×) |
| | | Non-sampling | 71.0 | 4368(1.00×) |
| | CORAL | $level_0$ | 76.3 | 496(0.04×) |
| | | $level_1$ | 78.1 | 1142(0.10×) |
| | | $level_2$ | 78.4 | 6678(0.59×) |
| | | Non-sampling | 85.3 | 11361 (1.00×) |

**Multi-times sampling & non-sampling.** We further do comparison between lines sampling in each epoch and non-sampling line. In this experiment: $level_0$ randomly samples 5 domains in each epoch; $level_1$ train an `invDANN` model only once to get the one-shot similarity matrix among domains based on which select 5 domains in each epoch; $level_2$ further select different batches in each epoch compared to $level_1$. The training epochs are set to be 75 for DANN and 50 for CORAL for sampling lines while 15 and 10 respectively for the non-sampling line. For all lines, the deployed Neural Network is Resnet18; DANN utilize SGD optimizer with learning rate 0.2, weight decay 5e-4 and momentum 0.9; CORAL leverage Adam optimizer with learning rate 0.001 and weight decay 0. Table 8 shows the mean accuracy and wall-clock time of each line and we make the following observations.

**Time overhead of pairwise domain operations.** Notice the proportion of the costed wall-clock time of non-sampling line and $level_0$. The proportion of CORAL, a method work by pairwise operations between domains, is apparently larger than that of DANN. Sampling($level_0$, $level_1$) conspicuously reduce the time overhead of CORAL.

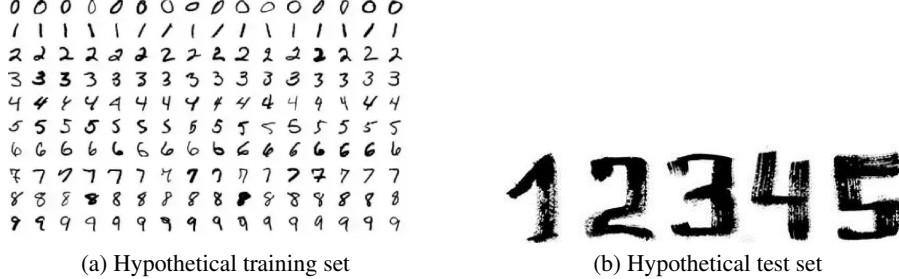

(a) Hypothetical training set    (b) Hypothetical test set

Figure 10: Two figures to illustrate the impact of object-induced spurious correlations on MNIST.

**The efficiency and drawbacks of `DOMI`.**    For one-shot sampling lines, `DOMI` substantially improves over random sampling at a acceptable extra overhead. Nevertheless, one-shot sampling leads to a large gap from non-sampling line since the model is consistently trained on the same small subset. For multi-times sampling lines, using the one-shot similarity matrix between domains, $level_1$ improve the performance over random sampling at a relatively small cost. However, $level_2$ need to sample batches from different subset of domains in each epoch, which means $level_2$ has to compute the similarity matrix between batches in each epoch, leading to great time overhead.

**Discussions**    Combined with Appendix B.3, it would be a significant future work to tackle two issues: extremely imbalanced data; computational overhead for algorithms that need to do sampling for multi-times. Moreover, parameter sharing between the trained model and the model used to sample is likely to be a practical means of reducing computational overhead.

## C    HOW CAN SPURIOUS CORRELATIONS OCCUR IN THE TWO DATASETS?

It's much easier to differentiate the rotation degree than discriminating the objects.

This can be empirically verified since it only needs about 30 epochs for a model to achieve over 98% validation accuracy of classifying 61 different degrees while 50 epochs to achieve no more than 97% and 88% validation accuracy of classifying 10 different objects on rotated MNIST and Fashion MNIST, respectively. Thus if a certain class label is closely associated with a certain rotation degree in the training set, recognizing objects by actually recognizing rotation degree can be a shortcut and domain-induced spurious correlation, just like classifying cats and lions using the background in the toy example. As for object-induced spurious correlation, on rotated MNIST, the handwriting is the feature of the object, however, it can also be the spurious correlation. For example, in Figure 10, let's focus on the number "1" and "7". After training on Figure 10a, can the model correctly recognize "1" in Figure 10b instead of wrongly taking it as "7" ?

On Fashion MNIST, assume we take the data in Figure 11 as the training set. Majority of the data points of Shirt are darker than Coat. When differentiate between Shirt and Coat, a model can simply

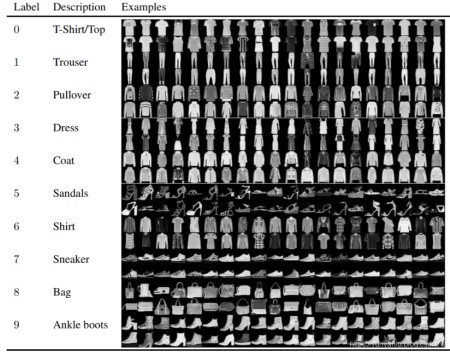

Figure 11: The figure to illustrate the impact of object-induced spurious correlations on Fashion MNIST.

takes the bright ones as coats and the dark ones as shirts to obtain high training accuracy. However, what if the color of Shirt and Coat is similar in the test set?

Thus our proposed framework set two levels of sampling to mitigate the impacts of domain-side and object-side, the sampling is in fact a rebalance procedure of data.

