# OpenReview forum: "Diversity Boosted Learning for Domain Generalization with a Large Number of Domains"
_ICLR.cc/2023/Conference — Submitted to ICLR 2023_

### Official Review · Reviewer_3y4o · 2022-10-25

**Confidence:** 3
**Correctness:** 3
**Technical Novelty And Significance:** 3
**Empirical Novelty And Significance:** 3
**Recommendation:** 5

**Clarity, Quality, Novelty And Reproducibility:**

## Clarity (W1)

Overall, the paper is written reasonably well. The one issue I found is that he authors appear to define the _domain_ quite narrowly, as basically the backgrounds of the images. I believe what authors call _object_ could also be a part of domain in the general sense. In the toy example that the authors use, the color of the object (cat or lion) can and should also be a part of the domain label. My understanding of the distinction between the object and domain then is that the authors assume that the domain information is incomplete and doesn't cover all of the relevant spurious features.

Formally, I think I can object to the presentation is Section 4.3, where the authors suggest that Proposition 1 in [2] is wrong. However, this proposition is a formal statement that is proved by [2]. I believe the issue is again that the "object-side" spurious feature $\hat x$ distribution is not the same across the training and testing domains in the toy cat-lion example, the authors assume that for the test domain the distribution of the object color (tan) $\hat x$ conditioned on the class label $y$ will be different compared to the training domains. I believe this setup contradicts the assumptions of Proposition 1 in [2].

## Methodology (W2)

Methodologically, it wasn't clear to me why we need to train a domain featurization which is class independent with invDANN? In particular, why couldn't you just apply ERM on the domain labels? What is the role of class invariance?

Similarly, couldn't you completely remove the first step of the procedure and just do the following?
1. Train ERM on all data with no domain information
2. Use a DPP with the ERM featurization to produce a diverse sample of datapoints (and the corresponding domains)

Is there a reason for the added complexity of the two-step procedure?

Further, the authors ablate the need for the second stage (adding diversity within sampled domains) but not the first stage of the procedure. What is the performance if you sample the domains randomly, but then use the ERM-based DPP to sample diverse datapoints?

## Experiments (W3)

The experiments show promising results, but are quite limited currently.

The authors only report performance on the Rotated MNIST and Rotated FashionMNIST datasets. This choice is unclear to me:
- Both datasets are small-scale. Further, these are not the most standard benchmarks in domain generalization to the best of my knowledge.
- The method is motivated by having a large number of domains and a large number of datapoints in each domain. While the number of domains is fairly large (65), it is unclear if the quadratic cost of using all the domains would already be an issue at this scale.
- It is unclear what the second stage of the procedure is supposed to be doing in these datasets. What is the spurious correlation that you are trying to address within each domain? Or maybe I am missing the point of the second stage here.

Generally, I think these datasets would be reasonable _if_ they were a part of a larger evaluation, which also included more realistic datasets where the motivation for the method is more clear. In fact, in the Appendix the authors also consider the iWildCam dataset, although they mention a few issues with the results, e.g. large computational overhead compared to the baselines and only using the FISH base domain generalization method, which performs relatively poorly on this dataset. At the same time, it does seem like the authors achieve some improvement on this dataset with FISH.

**Strength And Weaknesses:**

## Strengths

**S1**: The proposed method makes intuitive sense.

**S2**: Empirical results show improvements compared to the baselines with multiple base domain generalization methods.

## Weaknesses

**W1**: I think some of the presentation regarding the "object spurious correlations" vs "domain spurious correlations" is somewhat imprecise

**W2**: Some of the design decisions are not very clearly motivated and not ablated, in particular the use of invDANN

**W3**: The Empirical evaluation is focusing primarily on Rotated MNIST and Rotated FashionMNIST, which are small-scale synthetic datasets, rather than more standard domain generalization datasets.

**Summary Of The Paper:**

The paper provides a new sampling strategy for domain and datapoint sampling in the domain generalization context. The method encourages sampling diverse domains and diverse datapoints within each domain. To achieve domain diversity, the authors train a class-adversarial neural network (the inverse of the DANN method in [1]) and use a Determinantal Point Process (DPP) with the features produced by this network. To achieve datapoint diversity, the authors train standard ERM on the datapoints sampled randomly from the sampled domains, and also use a DPP.

**Summary Of The Review:**

This is an interesting paper, with promising results. Above, I highlighted a few questions and concerns about the presentation, methodology and experiments. The limited empirical evaluation is the main issue in my opinion.

## References

[1] [_Domain-Adversarial Training of Neural Networks_](https://arxiv.org/abs/1505.07818);
Yaroslav Ganin, Evgeniya Ustinova, Hana Ajakan, Pascal Germain, Hugo Larochelle, François Laviolette, Mario Marchand, Victor Lempitsky

[2][_Domain Generalization using Causal Matching_](https://arxiv.org/abs/2006.07500);
Divyat Mahajan, Shruti Tople, Amit Sharma

---

> ### Author Response · Authors · 2022-11-11
> **Reply to Reviewer 3y4o**
>
>
> >### 5. The method is motivated by having a large number of domains and a large number of datapoints in each domain. While the number of domains is fairly large (65), it is unclear if the quadratic cost of using all the domains would already be an issue at this scale.
> Motivated by your comment, we have supplemented experiments to compare sampling and non-sampling lines including one-shot sampling and multi-time sampling and the results are as follows. The empirical setting and more discussions are listed in Appendix B.4 in the revised version.
> - **Comparison between multi-time sampling and non-sampling lines**
>
> |  Dataset  | Backbone Algorithm | Sampling scheme  |Accuracy  |Wall-clock Time  |
> |  :----------:  | :----:  | :----:  | :----:  | :----:  |
> |CIFAR10-C|DANN|$level_0$|82.2|923($0.22 \times$)|
> |CIFAR10-C|DANN|$level_1$| 83.5|1531($0.37 \times$)|
> |CIFAR10-C|DANN|$level_2$| 83.9|6712($1.61 \times$)|
> |CIFAR10-C|DANN|Non-sampling| 89.3|4164($1.00 \times$)|
> |CIFAR10-C|CORAL|$level_0$| 87.8|461($0.05 \times$)|
> |CIFAR10-C|CORAL|$level_1$| 89.2|1004($0.10 \times$)|
> |CIFAR10-C|CORAL|$level_2$| 89.7|6175($0.61  \times$)|
> |CIFAR10-C|CORAL|Non-sampling| 94.2|10152($1.00 \times$)|
> |CIFAR100-C|DANN|$level_0$|61.7|945($0.22 \times$)|
> |CIFAR100-C|DANN|$level_1$|  62.9|1674($0.38 \times$)|
> |CIFAR100-C|DANN|$level_2$| 63.4|6864($1.57 \times$)|
> |CIFAR100-C|DANN|Non-sampling| 71.0|4368($1.00 \times$)|
> |CIFAR100-C|CORAL|$level_0$| 76.3|496($0.04 \times$)|
> |CIFAR100-C|CORAL|$level_1$|78.1|1142($0.10 \times$)|
> |CIFAR100-C|CORAL|$level_2$| 78.4|6678($0.59 \times$)|
> |CIFAR100-C|CORAL|Non-sampling| 85.3|11361 ($1.00 \times$)|
>
> - **Comparison between one-shot sampling and non-sampling lines**
>
> |  Dataset  | Backbone Algorithm | Sampling scheme  |Accuracy  |Wall-clock Time  |
> |  :----------:  | :----:  | :----:  | :----:  | :----:  |
> |CIFAR10-C|DANN|$level_0$| 63.4|916($0.22 \times$)|
> |CIFAR10-C|DANN|$level_1$| 64.2|1325($0.32 \times$)|
> |CIFAR10-C|DANN|$level_2$| 64.6|1523($0.37 \times$)|
> |CIFAR10-C|DANN|Non-sampling| 89.3|4164($1.00 \times$)|
> |CIFAR10-C|CORAL|$level_0$| 68.5|461($0.05 \times$)|
> |CIFAR10-C|CORAL|$level_1$|  70.1|863($0.09 \times$)|
> |CIFAR10-C|CORAL|$level_2$| 70.7|1074($0.11 \times$)|
> |CIFAR10-C|CORAL|Non-sampling| 94.2|10152($1.00 \times$)|
> |CIFAR100-C|DANN|$level_0$|33.9|945($0.22 \times$)|
> |CIFAR100-C|DANN|$level_1$| 34.9|1427($0.33 \times$)|
> |CIFAR100-C|DANN|$level_2$| 35.3|1664($0.38 \times$)|
> |CIFAR100-C|DANN|Non-sampling| 71.0|4368($1.00 \times$)|
> |CIFAR100-C|CORAL|$level_0$| 35.2|496($0.04 \times$)|
> |CIFAR100-C|CORAL|$level_1$|36.3|902($0.08 \times$)|
> |CIFAR100-C|CORAL|$level_2$| 37.0|1158($0.10 \times$)|
> |CIFAR100-C|CORAL|Non-sampling| 85.3|11361 ($1.00 \times$)|
>
> The results show that **1)** The computation overhead of CORAL, a method work by pairwise operations between domains, is much greater than that of DANN. **2)** The efficiency and drawbacks of DOMI: For one-shot sampling lines, DOMI substantially improves over random sampling at an acceptable extra overhead. Nevertheless, one-shot sampling leads to a large gap from the non-sampling line since the model is consistently trained on the same small subset. For multi-times sampling lines, using the one-shot similarity matrix between domains, $level_1$ improves the performance over random sampling at a relatively small cost. However, $level_2$ needs to sample batches from different subsets of domains in each epoch, which means $level_2$ has to compute the similarity matrix between batches in each epoch, leading to great time overhead.
>
>
> We can see that the influence resulting from the increase of domains is much more significant on CORAL, a method working by pairwise domain operations than on DANN. Using all the domains may not be an issue at this scale, we just use these four datasets in the paper to show the performance of DOMI, but when faced with the circumstances such as datasets in WILDS and DrugOOD mentioned in the paper which contain tens of thousand domains or even more, using all domains to train is computationally prohibitive.
> >### 6. It is unclear what the second stage of the procedure is supposed to be doing in these datasets. What is the spurious correlation that you are trying to address within each domain? Or maybe I am missing the point of the second stage here.
>
> Thanks for your insightful reviews. We have a discussion in Appendix C of the original submission to express how spurious correlations can occur in the two MNIST datasets.
>
> Thanks very much again for your valuable comments on our work.
>
> [1] hendrycks/robustness: Corruption and Perturbation Robustness (ICLR 2019) (github.com)
> [2]Domain Generalization using Causal Matching; Divyat Mahajan, Shruti Tople, Amit Sharma

---

> ### Author Response · Authors · 2022-11-11
> **Reply to Reviewer 3y4o**
>
> Dear reviewer 3y4o,
> Thank you sincerely for your time and efforts in reviewing our paper and comprehensive and valuable comments. Please kindly find our responses to your raised questions below. We hope our response can address your concerns.
>
> >### 1. Some of the design decisions are not very clearly motivated and not ablated, in particular the use of invDANN & Methodologically, it wasn't clear to me why we need to train a domain featurization which is class independent with invDANN? In particular, why couldn't you just apply ERM on the domain labels? What is the role of class invariance?
>
> Thanks for your comment. Following your suggestion, we have added the theoretical analysis to the revised version to show that diverse domains help exclude spurious correlations. To select diverse domains, we use invDANN to learn domain-side features. Suppose we use a well-trained OOD model which only extracts the object-side features and the distribution of objects is IID, then selecting domains by DPP based on this model is no different from the random sampling. Compared with using ERM to train a featurizer to help select domains, invDANN can further leverage the domain label to learn more information of the domain. The training procedure of ERM contains no operation on domain labels and thus somehow a waste of data. We use ERM in the second level exactly because of the lack of available labels for spurious correlations caused by objects. Moreover, the invDANN promotes the sampling in the second level of DOMI. "ERM is prone to taking shortcuts and learning spurious correlations... Combining these two, the ERM model is prone to relying on object-induced spurious correlations and thus can extract their information".
>
> >### 2. The Empirical evaluation is focusing primarily on Rotated MNIST and Rotated FashionMNIST, which are small-scale synthetic datasets, rather than more standard domain generalization datasets.
>
> Thanks for your suggestion. Motivated by your comment, we have extended our experiments to another two tougher datasets(Cifar10-c and Cifar100-c)[1] and the main results are as follows. The detailed setting and more results are listed in Section 5.3 and Appendix B.2 in the revised version.
>
> - **Average test accuracy of three algorithms. We repeat the experiment for 20 times with random seeds.**
>
> |  Dataset  | Sampling scheme  | DANN  |MMD  |CORAL  |
> |  :----:  | :----:  | :----:  | :----:  | :----:  |
> | CIFAR10-C  | $level_0$ |63.4 |65.0 |68.5 |
> | CIFAR10-C  |  $level_1$ |64.2 |66.5 |70.1 |
> | CIFAR10-C  |  $level_2$ |**64.6** |**66.9** |**70.7** |
> | CIFAR100-C  | $level_0$ |33.9 |33.8 |35.1 |
> | CIFAR100-C  |  $level_1$ |34.9 |35.2 |36.3 |
> | CIFAR100-C  |  $level_2$ |**35.3** |**35.7** |**37.0** |
>
> The results show the improvements of DOMI over the baseline($level_0$) across datasets and backbone algorithms.
>
> >### 3. The one issue I found is that he authors appear to define the domain quite narrowly, as basically the backgrounds of the images. I believe what authors call object could also be a part of domain in the general sense
>
> Thank you again for your constructive comments and for widening our understanding of the domain, our work can be a branch under work taking the unlabeled domain information into consideration in DG field and circumstances where the information of objects which can lead to spurious correlations is unlabeled is not rare. For example, the iwildcam dataset in WILDS benchmark only labels the location where the photos are caught, lacks the labels of the wild animals, similar in WaterBirds dataset.
>
> >### 4. Formally, I think I can object to the presentation is Section 4.3, where the authors suggest that Proposition 1 in [2] is wrong.
>
> Sorry for the misunderstanding that might be caused by our presentation. First we didn’t state that Proposition 1 in [2] is wrong. What we want to express is: [2] proposes under the condition that the distribution of “stable” features given $Y$ ($P(x_c \mid Y)$) is variant across domains, correlations satisfying independent of domains given $Y$ ($\Phi(x) ⫫ D \mid Y$) are not necessarily causal correlations, we further propose this conclusion without the condition. Moreover, we in fact treat the body color of objects as a spurious feature rather than a “stable” feature in our structure causal model. From another point of view, we separate the “stable” feature into two parts according to whether their distribution changes across domains and treat the changing part as spurious features.

---

> ### Author Response · Authors · 2022-11-16
> **Look forward to your feedback!**
>
> Dear Reviewer 3y4o,
>
> We sincerely thank you for your time and efforts in reviewing our paper, and appreciate your detailed and constructive comments. We have carefully revised the manuscript by incorporating your suggestions.
> Since the discussion time is ending soon, we would like to kindly remind you to check our responses and the revised version of paper. We hope it can address your concerns and look forward to your feedback.

---

> ### Author Response · Authors · 2022-12-10
> **To Reviewer 3y4o**
>
> Dear Reviewer 3y4o,
> We cherish your constructive comments and helpful suggestions. Since the time for review is ending soon, we would like to kindly remind you to check our responses and the revised version of our paper. We hope they can address your concerns and look forward to your feedback.

---

### Official Review · Reviewer_eKSv · 2022-10-25

**Confidence:** 3
**Clarity, Quality, Novelty And Reproducibility:** Please see the comments before.
**Correctness:** 2
**Technical Novelty And Significance:** 3
**Empirical Novelty And Significance:** 3
**Recommendation:** 6

**Strength And Weaknesses:**

Strengths
The problem of DG considered and the issues highlighted such as scalability and objective in this work is very relevant.
The authors provide an important counter-example, i.e., cats and lions to draw attention to the issue of data-side impacts. They also present the comparison of various sampling methods such as random sampling, and different diversity-boosted sampling methods to illustrate the importance of their work and their effect on both object-side spurious correlations and domain-side spurious correlations.

The authors give reasonable arguments for observation 1 using concepts such as good, casual, and spurious correlations. In addition, observation 2 is well explained using the structural causal model.
The solution proposed, i.e., diversity-boosted two-level sampling to mitigate the issues highlighted seems intuitively well grounded.
The experimental results show the validity of their algorithm. The proposed method outperforms level 0 and level 1 sampling schemes across 5 backbone DG methods and two simulated benchmark datasets.
The authors provide a decent analysis of the gap between test accuracy and maximal accuracy as well as the impact of the choice of the hyper-
parameter δ (proportion of data points) .

Drawbacks:

No analysis regarding the choice of the hyper-parameter β (proportion of domains) is presented.
The statement “spurious correlations essentially result from imbalance data” is not entirely true. The spurious correlations result from the existence of anti-causal paths. You could have the same data and different causal models, some of which result in spurious correlations.
It would be better if the authors elaborate on causality-related concepts such as causal correlations and unobserved confounders or give proper references.
There are a couple of typos (is composed by Featurizer, Classifier ..., while the test accuracy of level 0 .... that of level 2 centers, and ...)
The experiments on large-scale datasets are missing.

**Summary Of The Paper:**

Summary
The authors consider the problem of domain generalization (DG). In specific, they are addressing the issues of scalability and objective.
 Scalability: The state-of-the-art DG methods involve computational
 complexity of 𝒪 (n2) corresponding to pairwise domain operations with n
domains. In addition, each domain may contain a large number of data points. This is computationally prohibitive.
Objective: The existing methods entirely focus on excluding domain-side impacts, i.e., spurious correlations, and don’t consider the data-side impacts.
The authors address the above two issues by proposing a two-level sampling method called the Diversity boosted two-level sampling framework (DOMI).
The authors argue that diverse domains of data help exclude spurious correlation (observation 1). In regard to this, they propose to sample diverse domains using invDANN as featurizer and DPP as a sampler in the first level.
The authors observe excluding domain-induced correlations is insufficient for learning a robust model (observation 2). To alleviate this, they propose to sample diverse data batches from the selected domains (domains obtained from first-level sampling) using ERM and DPP in the second level.
The authors validate their algorithm with five backbone DG algorithms on two simulated benchmarks (Rotated MNIST and Rotated Fashion MNIST).

**Summary Of The Review:**

The authors consider the problem of domain generalisation (DG). In specific, they are addressing the issues of scalability and objective.

---

> ### Author Response · Authors · 2022-11-11
> **Reply to Reviewer eKSv**
>
> Dear reviewer eKSv,
>
> Thanks for your time and efforts in reviewing our paper. We have uploaded the new version of paper and here is our detailed response below. We hope our response can address your concerns:
> >### 1. The statement “spurious correlations essentially result from imbalance data” is not entirely true. The spurious correlations result from the existence of anti-causal paths. You could have the same data and different causal models, some of which result in spurious correlations.
>
> This statement indeed can be improved as you suggested, thanks again for your efforts and valuable comments to let us be able to correct our description and make it clearer.
> We update this statement as *“Under the circumstance of imbalanced data where specific clusters contain the majority of data points while the others are only a tiny fraction of the training set, there likely to exist an anti-causal path, i.e., a spurious correlation only catching some properties of the large clusters of the data, and algorithms minimizing the average loss like ERM may simply take this correlation as the causal correlation”* in Section 3 of the revised paper. We further use an example on chest x-ray dataset to help explain and support the statement.
> >### 2. No analysis regarding the choice of the hyper-parameter β (proportion of domains) is presented.
>
> We indeed lack this analysis since a larger β, i.e., sampling more domains to train on will undoubtedly enhance the performance of the model while bring greater computational overhead, especially for methods working by pairwise operations between domains at the same time. We show the efficiency and performance of DOMI, and users need to make a tradeoff on the choice of β when using DOMI according to their computational resources and the goal to achieve.
>
> >### 3.The experiments on large-scale datasets are missing.
>
> Inspired by your comment, we have conducted more experiments to evaluate DOMI in the new version of paper. Specifically, we extend our experiments to another two tougher datasets, CIFAR10-C and CIFAR100-C[1]. The main results are as follows. We further conduct experiments to compare sampling and non-sampling lines. The detailed setting and results are listed in Section 5.3, Appendix B.2 and Appendix B.4 in the revised version of paper.
> - **Average test accuracy of three algorithms. We repeat the experiment for 20 times with random seeds.**
>
> |  Dataset  | Sampling scheme  | DANN  |MMD  |CORAL  |
> |  :----:  | :----:  | :----:  | :----:  | :----:  |
> | CIFAR10-C  | $level_0$ |63.4 |65.0 |68.5 |
> | CIFAR10-C  |  $level_1$ |64.2 |66.5 |70.1 |
> | CIFAR10-C  |  $level_2$ |**64.6** |**66.9** |**70.7** |
> | CIFAR100-C  | $level_0$ |33.9 |33.8 |35.1 |
> | CIFAR100-C  |  $level_1$ |34.9 |35.2 |36.3 |
> | CIFAR100-C  |  $level_2$ |**35.3** |**35.7** |**37.0** |
>
> DOMI substantially improves over the baseline($level_0$) across datasets and backbone algorithms.
>
> >### 4. There are a couple of typos (is composed by Featurizer, Classifier ..., while the test accuracy of level 0 .... that of level 2 centers, and ...)
>
> Thank you sincerely for your efforts in reviewing our paper, we verified the spelling carefully and uploaded the new version of paper.
>
> We’d appreciate your patience and welcome any further discussions or questions!
> [1] hendrycks/robustness: Corruption and Perturbation Robustness (ICLR 2019) (github.com)

---

> > ### Comment · Reviewer_eKSv · 2022-11-16
> > **Response on the Revision.**
> >
> > I thank the Authors for incorporating some of my feedback.
> >
> > I belive that this paper is slightly above the acceptance threshold and thus keep my ratings intact.

---

> > > ### Author Response · Authors · 2022-11-17
> > > **Reply to Reviewer eKSv**
> > >
> > > Dear Reviewer eKSv,
> > >
> > > Thank you again for your efforts and suggestions, from which we benefited a lot.

---

> ### Author Response · Authors · 2022-11-16
> **Look forward to your feedback!**
>
> Dear Reviewer eKSv,
>
> We sincerely thank you for your time and efforts in reviewing our paper, and appreciate your detailed and constructive comments. We have carefully revised the manuscript by incorporating your suggestions.
> Since the discussion time is ending soon, we would like to kindly remind you to check our responses and the revised version of paper. We hope it can address your concerns and look forward to your feedback.

---

### Official Review · Reviewer_kTC5 · 2022-11-02

**Confidence:** 5
**Correctness:** 2
**Technical Novelty And Significance:** 1
**Empirical Novelty And Significance:** Not applicable
**Recommendation:** 5

**Clarity, Quality, Novelty And Reproducibility:**

Clarity: excellent, easy to follow, well explained
Novelty: unsure. While the sampling methods seems to be working well compared to other sampling, it is not the contribution of the paper since they use the existing DPP.
Reproducibility: good

**Strength And Weaknesses:**

Pros:
- good observations (spurious correlations)
- interesting idea to choose diverse domains and data points
- clarity - pretty easy to follow
Cons:
- overly complicated multi-stage method that requires many expensive computations (train inverse DANN, construct matrix of pairwise domain similarity, construct matrix of data points similarities, train ERM etc)
- no comparison with no sampling whatsoever - so impossible to say if it was needed in the first place
- only image experiments. Is it because you can't find non image DG datasets with such a large number of domains? is it possible to generate some synthetic data? why not to try WILDS and OGB-MolPCBA that you mention? Or DRUG OOD

**Summary Of The Paper:**

Authors looks into Domain Generalization for a large number of domains. They propose to sample (using two stage procedure) both the domains and the data within each domain, in order to help the model to filter out spurious correlation.  They show a motivational example that the assumption that domains usually have shared causal features and differ in spurious correlation features does not always hold. For the first stage sampling (choosing the domains), they  train an inverse DANN (classify domains while not being able to assign the class labels correctly) to get to extract only domain specific features. Then using these features, they construct domain similarity matrix . For the second stage sampling,  DPP samples diverse domains and diverse data points within the domains (using similarity matrix constructed by ERM)


**Summary Of The Review:**

Update:
Raising my score a bit, since authors included new experiments. TLDR: without subsampling the performance does improve but on some datasets the drop with sampling is small enough considered time saved


Overall my main complains about this paper are
1) multi step procedure with each step step being expensive (train inverse DANN, construct matrix of domain similariities, construct data points similarities matrix etc). Each step will require its own tuning. Unclear how your method compares in terms of time/computation to any methods you labeled as expensive
2) No comparison with no sampling at all for the table 3. Without this, it is impossible to tell whether the sampling is required at all
3) You state that DANN works with pairwise domains (n^2)??? You have an adversarial head that classifies into the number of domains, how is it n^2??? Also it is misleading to state this for  MMD too - it will indeed work on pairs, but only on pairs from the batch, so it is batch_size^2 max

Minor: DPP related work - repulsive (interactions) is a strange word to use here. Spurious? Non intended?

---

> ### Author Response · Authors · 2022-11-11
> **Response to Reviewer kTC5**
>
>
> >### 3. only image experiments. Is it because you can't find non image DG datasets with such a large number of domains? is it possible to generate some synthetic data? why not to try WILDS and OGB-MolPCBA that you mention? Or DRUGOOD
>
> Thanks for your suggestion. We have extended our experiments to two more challenging datasets(Cifar10-c and Cifar100-c)[1] to give further support to DOMI and the results are as follows. More results and discussions are listed in Section 5.3 in the new version of paper. Moreover, DOMI can be easily extended to non-image data using corresponding encoders, such as transformer for text data and GNN for graph data, and we are working on non-image datasets in our future work.
>
> - **Average test accuracy of three algorithms. We repeat the experiment for 20 times with random seeds.**
>
> |  Dataset  | Sampling scheme  | DANN  |MMD  |CORAL  |
> |  :----:  | :----:  | :----:  | :----:  | :----:  |
> | CIFAR10-C  | $level_0$ |63.4 |65.0 |68.5 |
> | CIFAR10-C  |  $level_1$ |64.2 |66.5 |70.1 |
> | CIFAR10-C  |  $level_2$ |**64.6** |**66.9** |**70.7** |
> | CIFAR100-C  | $level_0$ |33.9 |33.8 |35.1 |
> | CIFAR100-C  |  $level_1$ |34.9 |35.2 |36.3 |
> | CIFAR100-C  |  $level_2$ |**35.3** |**35.7** |**37.0** |
>
> DOMI substantially improves over the baseline($level_0$) across datasets and backbone algorithms.
>
> >### 4. You state that DANN works with pairwise domains (n^2)??? You have an adversarial head that classifies into the number of domains, how is it n^2??? Also it is misleading to state this for MMD too - it will indeed work on pairs, but only on pairs from the batch, so it is batch_size^2 max
>
> Thanks for your comment. We described the computational overhead of the five backbone algorithms in the original submission as follows: "Discussions. Although MatchDG, FISH, MMD and CORAL perform well in domain generalization tasks, the matching procedure between domains means their computational complexity is O($n^2$) with n training domains."(Section 2); "Backbones. We take MatchDG (Mahajan et al., 2021), FISH (Shi et al., 2021b), CORAL (Sun & Saenko, 2016), MMD (Li et al., 2018b) and DANN (Ganin et al., 2016) as backbone algorithms. The former four algorithms work by pairwise domain operations, leading to O($n^2$) computational complexity with n domains" (Section 5.1). Thus, we only state that MatchDG, FISH, MMD and CORAL work by pairwise operations between domains. Moreover, MMD and CORAL work on pairs from the batch in each round and the computation overhead is in expectation O($n^2$) with n domains.
>
> >### 5. Minor: DPP related work - repulsive (interactions) is a strange word to use here. Spurious? Non intended?
>
> Sorry for the misunderstanding. “Repulsive” here has nothing to do with “Spurious” or ”Non intended”, just says that the samples selected by DPP are dissimilar. As is used in published papers, “repulsive” is indeed the right word for DPP: “The DPP is a point process which mimics repulsive interactions between samples”[2].
>
> Thank you again for your efforts in reviewing our paper. We hope our response can address your concerns.
>
>
> [1] hendrycks/robustness: Corruption and Perturbation Robustness (ICLR 2019) (github.com)
> [2] Zhang C, Kjellstrom H, Mandt S. Determinantal point processes for mini-batch diversification[J]. arXiv preprint arXiv:1705.00607, 2017.

---

> > ### Comment · Reviewer_kTC5 · 2022-11-28
> > **additional comments**
> >
> > Thanks for additional experiments
> > - Are the numbers/differences you highlight significant (e.g. 66.5 vs 66.9)?
> > - I disagree about DANN computational complexity. It is just a head (dimension n) - it does not work on n^2 pairs of domains.
> > Thanks

---

> ### Author Response · Authors · 2022-11-11
> **Response to Reviewer kTC5**
>
> Dear reviewer kTC5,
>
> Thanks for your insightful reviews, and we appreciate the valuable suggestions! We’ve revised the manuscript and added additional experiments according to your suggestions. Please kindly find our response to your raised questions below.
> >### 1. No comparison with no sampling whatsoever - so impossible to say if it was needed in the first place
>
> Inspired by your comment, we have supplemented experiments to compare sampling and non-sampling lines including one-shot sampling and multi-time sampling in the revised version. We record the mean accuracy together with wall-clock time and here are the results. More analysis as well as discussions are listed in Appendix B.4 in the revised version of paper.
>
> - **Comparison between multi-times sampling and non-sampling lines**
>
> |  Dataset  | Backbone Algorithm | Sampling scheme  |Accuracy  |Wall-clock Time  |
> |  :----------:  | :----:  | :----:  | :----:  | :----:  |
> |CIFAR10-C|DANN|$level_0$|82.2|923($0.22 \times$)|
> |CIFAR10-C|DANN|$level_1$| 83.5|1531($0.37 \times$)|
> |CIFAR10-C|DANN|$level_2$| 83.9|6712($1.61 \times$)|
> |CIFAR10-C|DANN|Non-sampling| 89.3|4164($1.00 \times$)|
> |CIFAR10-C|CORAL|$level_0$| 87.8|461($0.05 \times$)|
> |CIFAR10-C|CORAL|$level_1$| 89.2|1004($0.10 \times$)|
> |CIFAR10-C|CORAL|$level_2$| 89.7|6175($0.61  \times$)|
> |CIFAR10-C|CORAL|Non-sampling| 94.2|10152($1.00 \times$)|
> |CIFAR100-C|DANN|$level_0$|61.7|945($0.22 \times$)|
> |CIFAR100-C|DANN|$level_1$|  62.9|1674($0.38 \times$)|
> |CIFAR100-C|DANN|$level_2$| 63.4|6864($1.57 \times$)|
> |CIFAR100-C|DANN|Non-sampling| 71.0|4368($1.00 \times$)|
> |CIFAR100-C|CORAL|$level_0$| 76.3|496($0.04 \times$)|
> |CIFAR100-C|CORAL|$level_1$|78.1|1142($0.10 \times$)|
> |CIFAR100-C|CORAL|$level_2$| 78.4|6678($0.59 \times$)|
> |CIFAR100-C|CORAL|Non-sampling| 85.3|11361 ($1.00 \times$)|
>
> - **Comparison between one-shot sampling and non-sampling lines**
>
> |  Dataset  | Backbone Algorithm | Sampling scheme  |Accuracy  |Wall-clock Time  |
> |  :----------:  | :----:  | :----:  | :----:  | :----:  |
> |CIFAR10-C|DANN|$level_0$| 63.4|916($0.22 \times$)|
> |CIFAR10-C|DANN|$level_1$| 64.2|1325($0.32 \times$)|
> |CIFAR10-C|DANN|$level_2$| 64.6|1523($0.37 \times$)|
> |CIFAR10-C|DANN|Non-sampling| 89.3|4164($1.00 \times$)|
> |CIFAR10-C|CORAL|$level_0$| 68.5|461($0.05 \times$)|
> |CIFAR10-C|CORAL|$level_1$|  70.1|863($0.09 \times$)|
> |CIFAR10-C|CORAL|$level_2$| 70.7|1074($0.11 \times$)|
> |CIFAR10-C|CORAL|Non-sampling| 94.2|10152($1.00 \times$)|
> |CIFAR100-C|DANN|$level_0$|33.9|945($0.22 \times$)|
> |CIFAR100-C|DANN|$level_1$| 34.9|1427($0.33 \times$)|
> |CIFAR100-C|DANN|$level_2$| 35.3|1664($0.38 \times$)|
> |CIFAR100-C|DANN|Non-sampling| 71.0|4368($1.00 \times$)|
> |CIFAR100-C|CORAL|$level_0$| 35.2|496($0.04 \times$)|
> |CIFAR100-C|CORAL|$level_1$|36.3|902($0.08 \times$)|
> |CIFAR100-C|CORAL|$level_2$| 37.0|1158($0.10 \times$)|
> |CIFAR100-C|CORAL|Non-sampling| 85.3|11361 ($1.00 \times$)|
>
> >### 2. multi step procedure with each step step being expensive (train inverse DANN, construct matrix of domain similariities, construct data points similarities matrix etc). Each step will require its own tuning. Unclear how your method compares in terms of time/computation to any methods you labeled as expensive
>
> Thanks for your insightful reviews. The supplemented experiments above comparing sampling and non-sampling lines show that $level_1$ improves over $level_0$ at a relatively small cost by using a one-shot similarity matrix between domains. When we need multi-time sampling, $level_2$ leads to great computational overhead. It would be a significant future work to tackle this issue and we hold that using the architecture with shared parameters between the trained model and the model used to sample can be a practical way to reduce computational cost.

---

> > ### Comment · Reviewer_kTC5 · 2022-11-28
> > **Thanks for additional experiments**
> >
> > Dear authors. Thank you for adding non sampling baseline. Just to make sure we are on the same page-  no sampling is basically training whatever method (DANN, Coral etc) on the full data (all the domains, all the data from all the domains) correct?
> > - If yes, it does seem that just doing non sampling always is better than any sampling (albeit computationally intensive).
> > - It seems that level 2 sampling is just as computationally expensive as no sampling.
> > - does wall time for level_i sampling includes the tuning of hyperparameters for your method? What about tuning for non sampling? Or is it just the run time you compare
> > Thanks

---

> > > ### Author Response · Authors · 2022-11-30
> > > **Thank you for your feedback**
> > >
> > > Thank you for your time in reviewing our paper and we are happy to receive your feedback! The non-sampling line is training on the full data, i.e., all the data from all the domains in each epoch whatever method.
> > > >### Does wall time for level_i sampling includes the tuning of hyperparameters for your method? What about tuning for non sampling? Or is it just the run time you compare
> > >
> > > We just compare the running time of the sampling lines and the non-sampling line. The critical extra hyperparameter for DOMI to be tuned is $\delta$ in $level_2$ and we set it as 0.9.
> > >
> > > >### It does seem that just doing non sampling always is better than any sampling (albeit computationally intensive).
> > >
> > > - The gap between multi-time sampling and non-sampling is not that large since we only use 5/51 domains for training in each epoch for the sampling lines. The gap would be further shrunk with a larger subset of domains.
> > > - Using all the domains may not be an issue at this scale (61 on MNISTs and 51 on CIFARs), we just use these four datasets in the paper to show the performance of DOMI compared with random sampling, but when faced with the circumstances such as datasets in WILDS and DrugOOD mentioned in the paper which contain tens of thousand domains or even more, using all domains to train is computationally prohibitive, especially for a method working by pairwise domain operations (the empirical results on the wall-clock time show that the influence resulting from the increase of domains is much more significant on CORAL than DANN), where sampling is necessary.
> > >
> > > >### It seems that level 2 sampling is just as computationally expensive as no sampling.
> > >
> > >
> > > For one-shot sampling lines, $level_2$ substantially improves over random sampling and $level_1$ at an acceptable extra overhead, which shows its efficacy. Nevertheless, one-shot sampling leads to a large gap from the non-sampling line since the model is consistently trained on the same small subset. For multi-times sampling lines, using the one-shot similarity matrix between domains, $level_1$ improves the performance over random sampling at a relatively small cost while $level_2$ needs to sample batches from different subsets of domains in each epoch, which means $level_2$ has to compute the similarity matrix between batches in each epoch, leading to great time overhead. In conclusion:
> > > - $level_2$ shows its efficacy in improving the robustness of distribution shifts
> > > - $level_1$ substantially outperforms random sampling at an acceptable extra overhead in both one-shot and multi-time settings.
> > > - Parameter sharing may be a practical method to reduce the extra computational overhead, and we will leave it as important future work.
> > >
> > >
> > > >### Are the numbers/differences you highlight significant (e.g. 66.5 vs 66.9)?
> > >
> > > Thanks for your question. The difference we highlight is relatively significant with CORAL while not that large with another two backbone algorithms with $\delta = 0.9$ which is not meticulously tuned. A possible way to boost the difference is leveraging the similarity matrix to adaptively decide the value of $\delta$ instead of adjusting it manually, e.g., $\delta$ should take a relatively small value when the data tend to be similar.
> > >
> > > >### I disagree about DANN computational complexity. It is just a head (dimension n) - it does not work on n^2 pairs of domains.
> > >
> > > We also agree that the computational complexity of DANN is not $O(n^2)$. We would like to clarify that in our original submission we did not state that the computational complexity of DANN is $O(n^2)$. We described the computational overhead of the five backbone algorithms in the original submission as follows: “Although MatchDG, FISH, MMD and CORAL perform well in domain generalization tasks, the matching procedure between domains means their computational complexity is $O(n^2)$ with n training domains.”(Section 2); “We take MatchDG (Mahajan et al., 2021), FISH (Shi et al., 2021b), CORAL (Sun & Saenko, 2016), MMD (Li et al., 2018b) and DANN (Ganin et al., 2016) as backbone algorithms. The former four algorithms work by pairwise domain operations, leading to $O(n^2)$ computational complexity with n domains” (Section 5.1).

---

> > > > ### Comment · Reviewer_kTC5 · 2022-12-12
> > > > **Thanks for the comments**
> > > >
> > > > Just to clarify - even MMD is not going to be O(n^2) - it can be max O(b^2) where b is the batch size (if you have a batch of 128 examples but you have 900 domains, you are not going to try all 900 domains for mmd pairs)

---

> > > > > ### Author Response · Authors · 2022-12-13
> > > > > **About the computational complexity of MMD**
> > > > >
> > > > > Thank you for raising our score. We would like to supplement some discussion on the complexity of MMD. MMD is essentially an OOD method working by domain alignment. It algorithmically calculate the distance between each pairs of domains and minimize it to ensure the learned representation is invariant across domains. Although in the code implementation MMD calculates the distance between domains in a mini-batch, it requires in expectation $O(n^2)$ operations to exert its effect. A simple example is that the data in a mini-batch cannot all come from the same domain, and when they come from different domains multiple epochs are required to achieve convergence. Meanwhile,  your comments suggest a promising direction for improving it in the mini-batch implementation: efficiently select the most informative data to form the mini-batch for accelerating the convergence (fewer epochs and wall-clock time).
> > > > > Thank you again for your constructive comments that helps improve our work a lot.

---

> ### Author Response · Authors · 2022-11-16
> **Look forward to your feedback!**
>
> Dear Reviewer kTC5,
>
> We sincerely thank you for your time and efforts in reviewing our paper, and appreciate your detailed and constructive comments. We have carefully revised the manuscript by incorporating your suggestions. Since the discussion time is ending soon, we would like to kindly remind you to check our responses and the revised version of paper. We hope it can address your concerns and look forward to your feedback.

---

### Official Review · Reviewer_xNqP · 2022-11-04

**Confidence:** 4
**Correctness:** 3
**Technical Novelty And Significance:** 2
**Empirical Novelty And Significance:** 2
**Recommendation:** 3

**Clarity, Quality, Novelty And Reproducibility:**

* Please see above. The paper can be more professionally and well written. The idea is new and promising but needs more support.
* Source code is provided.

**Strength And Weaknesses:**

The paper is easy to follow and the proposed algorithm is easy to understand. The idea is not totally new but seems promising. Since the work is likely considered as adopting and improving existing method, I believe the experimental results should be strong. However, I am not fully convinced by the current experiments. Here are my complaints:
* The paper can be more professionally written. For example, the definition of set $C$ is very vague. What is precisely a "good" correlation?
* The experiments are very limited. Why not try some large scale experiments mentioned in DomainBed? It is important to show the proposed algorithm on larger and tougher dataset. I am also curious to see how it works on ColoredMNIST.


**Summary Of The Paper:**

This paper:
* solves the domain generalization problem
* presents several observations that diversity helps mitigate serious correlations
* proposes a sampling method that helps train robust models
* conducts experiment on Rotated MNIST and Rotated Fashion MNIST to show the effectiveness of the proposed algorithm

**Summary Of The Review:**

I think the paper is not ready for publication at this point. If the authors can show more evidence and support, I am happy to raise my scores.

---

> ### Author Response · Authors · 2022-11-11
> **Response to Reviewer xNqP**
>
>
> >### 2.The experiments are very limited. Why not try some large scale experiments mentioned in DomainBed? It is important to show the proposed algorithm on larger and tougher dataset. I am also curious to see how it works on ColoredMNIST.
>
> Inspired by your comment, we have conducted more experiments for evaluation in the revised version. Since the standard colored MNIST dataset in DomainBed benchmark only holds 3 domains in total, it may not fit for the setting of a large number of domains. We extend our experiments to another two tougher datasets(CIFAR10-C and CIFAR100-C)[1], of which the data are colored images instead of greyscale ones and generated by various corruption functions with different severity to add noises such as frog and frost effects. We list the main results here. The detailed setting and more results can be found in Section 5.3 and Appendix B.2 in the new version of paper.
>
> - **Average test accuracy of three algorithms. We repeat the experiment for 20 times with random seeds.**
>
> |  Dataset  | Sampling scheme  | DANN  |MMD  |CORAL  |
> |  :----:  | :----:  | :----:  | :----:  | :----:  |
> | CIFAR10-C  | $level_0$ |63.4 |65.0 |68.5 |
> | CIFAR10-C  |  $level_1$ |64.2 |66.5 |70.1 |
> | CIFAR10-C  |  $level_2$ |**64.6** |**66.9** |**70.7** |
> | CIFAR100-C  | $level_0$ |33.9 |33.8 |35.1 |
> | CIFAR100-C  |  $level_1$ |34.9 |35.2 |36.3 |
> | CIFAR100-C  |  $level_2$ |**35.3** |**35.7** |**37.0** |
>
> One can observe that DOMI substantially improves over the baseline ($level_0$) across datasets and backbone algorithms. We further conduct experiments to compare sampling and non-sampling lines.
> - **Comparison between multi-time sampling and non-sampling lines**
>
> |  Dataset  | Backbone Algorithm | Sampling scheme  |Accuracy  |Wall-clock Time  |
> |  :----------:  | :----:  | :----:  | :----:  | :----:  |
> |CIFAR10-C|DANN|$level_0$|82.2|923($0.22 \times$)|
> |CIFAR10-C|DANN|$level_1$| 83.5|1531($0.37 \times$)|
> |CIFAR10-C|DANN|$level_2$| 83.9|6712($1.61 \times$)|
> |CIFAR10-C|DANN|Non-sampling| 89.3|4164($1.00 \times$)|
> |CIFAR10-C|CORAL|$level_0$| 87.8|461($0.05 \times$)|
> |CIFAR10-C|CORAL|$level_1$| 89.2|1004($0.10 \times$)|
> |CIFAR10-C|CORAL|$level_2$| 89.7|6175($0.61  \times$)|
> |CIFAR10-C|CORAL|Non-sampling| 94.2|10152($1.00 \times$)|
> |CIFAR100-C|DANN|$level_0$|61.7|945($0.22 \times$)|
> |CIFAR100-C|DANN|$level_1$|  62.9|1674($0.38 \times$)|
> |CIFAR100-C|DANN|$level_2$| 63.4|6864($1.57 \times$)|
> |CIFAR100-C|DANN|Non-sampling| 71.0|4368($1.00 \times$)|
> |CIFAR100-C|CORAL|$level_0$| 76.3|496($0.04 \times$)|
> |CIFAR100-C|CORAL|$level_1$|78.1|1142($0.10 \times$)|
> |CIFAR100-C|CORAL|$level_2$| 78.4|6678($0.59 \times$)|
> |CIFAR100-C|CORAL|Non-sampling| 85.3|11361 ($1.00 \times$)|
>
> - **Comparison between one-shot sampling and non-sampling lines**
>
> |  Dataset  | Backbone Algorithm | Sampling scheme  |Accuracy  |Wall-clock Time  |
> |  :----------:  | :----:  | :----:  | :----:  | :----:  |
> |CIFAR10-C|DANN|$level_0$| 63.4|916($0.22 \times$)|
> |CIFAR10-C|DANN|$level_1$| 64.2|1325($0.32 \times$)|
> |CIFAR10-C|DANN|$level_2$| 64.6|1523($0.37 \times$)|
> |CIFAR10-C|DANN|Non-sampling| 89.3|4164($1.00 \times$)|
> |CIFAR10-C|CORAL|$level_0$| 68.5|461($0.05 \times$)|
> |CIFAR10-C|CORAL|$level_1$|  70.1|863($0.09 \times$)|
> |CIFAR10-C|CORAL|$level_2$| 70.7|1074($0.11 \times$)|
> |CIFAR10-C|CORAL|Non-sampling| 94.2|10152($1.00 \times$)|
> |CIFAR100-C|DANN|$level_0$|33.9|945($0.22 \times$)|
> |CIFAR100-C|DANN|$level_1$| 34.9|1427($0.33 \times$)|
> |CIFAR100-C|DANN|$level_2$| 35.3|1664($0.38 \times$)|
> |CIFAR100-C|DANN|Non-sampling| 71.0|4368($1.00 \times$)|
> |CIFAR100-C|CORAL|$level_0$| 35.2|496($0.04 \times$)|
> |CIFAR100-C|CORAL|$level_1$|36.3|902($0.08 \times$)|
> |CIFAR100-C|CORAL|$level_2$| 37.0|1158($0.10 \times$)|
> |CIFAR100-C|CORAL|Non-sampling| 85.3|11361 ($1.00 \times$)|
>
> The results show **1)** The computation overhead of CORAL, a method work by pairwise operations between domains, is much more significant than that of DANN. **2)** The efficiency and drawbacks of DOMI: For one-shot sampling lines, DOMI substantially improves over random sampling at an acceptable extra overhead. Nevertheless, one-shot sampling leads to a large gap from the non-sampling line since the model is consistently trained on the same small subset. For multi-time sampling lines, using the one-shot similarity matrix between domains, $level_1$ improves the performance over random sampling at a relatively small cost. However, $level_2$ needs to sample batches from different subsets of domains in each epoch, which means $level_2$ has to compute the similarity matrix between batches in each epoch, leading to great time overhead. Detailed experimental settings and more results are listed in Appendix B.4.
>
> Thank you again for your efforts in reviewing our paper. We’d be grateful if our response could address your concerns.
>
> [1]hendrycks/robustness: Corruption and Perturbation Robustness (ICLR 2019) (github.com)

---

> ### Author Response · Authors · 2022-11-11
> **Reply to Reviewer xNqP**
>
> Hello reviewer xNqP,
>
> We would like to thank you for your time spent reviewing our paper and providing constructive comments. Please kindly find our responses to your raised questions below:
> >### 1.The paper can be more professionally written. For example, the definition of set C is very vague. What is precisely a "good" correlation?
>
> Thanks for your suggestion. Motivated by your comment, we have formally defined the invariance set, correlation and diversity in Section 4.1, based on which we propose Proposition 1 to show diverse domains help mitigate spurious correlations in the revised version.

---

> ### Author Response · Authors · 2022-11-16
> **Look forward to your feedback!**
>
> Dear Reviewer xNqP,
>
> We sincerely thank you for your time and efforts in reviewing our paper, and appreciate your detailed and constructive comments. We have carefully revised the manuscript by incorporating your suggestions.
> Since the discussion time is ending soon, we would like to kindly remind you to check our responses and the revised version of paper. We hope it can address your concerns and look forward to your feedback.

---

> ### Author Response · Authors · 2022-12-10
> **To Reviewer xNqP**
>
> Dear reviewer xNqP,
> We cherish your constructive comments and helpful suggestions. Since the time for review is ending soon, we would like to kindly remind you to check our responses and the revised version of our paper. We hope they can address your concerns and look forward to your feedback.

---

### Author Response · Authors · 2022-11-11
**Reply to all**

We sincerely thank the reviewers for their efforts and valuable comments. We have revised our paper based on the suggestions of all four reviewers using blue lines. In detail:
- We have formally defined the correlation and diversity between domains, based on which we derive Proposition1: "Diverse domains help exclude spurious correlations" to give theoretical support to DOMI.
- We have extended our experiments to another two tougher datasets (CIFAR10-C and CIFAR100-C [1]).
- We have provided experimental results to compare sampling and non-sampling methods.

[1] hendrycks/robustness: Corruption and Perturbation Robustness (ICLR 2019) (github.com)

---

### Decision · Program_Chairs · 2023-01-20

**Decision:**

Reject

**Justification For Why Not Higher Score:**

The contribution of the work is below the acceptance bar.

**Justification For Why Not Lower Score:**

N/A

**Metareview: Summary, Strengths And Weaknesses:**

The paper is well-written and easy to follow. The observation is interesting and inspiring. The major concern is that the proposed method is a bit complicated and involves extra computing overhead. The performance gain achieved by the method is relatively small compared to the computing cost. In addition, the experiments are mainly performed on small datasets. Although the authors claim that the necessity of using sample-based methods for domain generalization is extremely high on large-scale datasets with multiple domains, no experiments are shown on these large-scale datasets, e.g., WILDS and DRUG OOD.